# The 3D mutational constraint on amino acid sites in the human proteome

Bian Li 🆔 [1,2✉], Dan M. Roden[2,3] & John A. Capra[1,4✉]

Quantification of the tolerance of protein sites to genetic variation has become a cornerstone of variant interpretation. We hypothesize that the constraint on missense variation at individual amino acid sites is largely shaped by direct interactions with 3D neighboring sites. To quantify this constraint, we introduce a framework called COntact Set MISsense tolerance (or COSMIS) and comprehensively map the landscape of 3D mutational constraint on 6.1 million amino acid sites covering 16,533 human proteins. We show that 3D mutational constraint is pervasive and that the level of constraint is strongly associated with disease relevance both at the site and the protein level. We demonstrate that COSMIS performs significantly better at variant interpretation tasks than other population-based constraint metrics while also providing structural insight into the functional roles of constrained sites. We anticipate that COSMIS will facilitate the interpretation of protein-coding variation in evolution and prioritization of sites for mechanistic investigation.

[1] Department of Biological Sciences, Vanderbilt University, Nashville, TN 37203, USA. [2] Department of Medicine, Vanderbilt University Medical Center, Nashville, TN 37232, USA. [3] Departments of Pharmacology and Biomedical Informatics, Vanderbilt University Medical Center, Nashville, TN 37232, USA. [4] Bakar Computational Health Sciences Institute and Department of Epidemiology and Biostatistics, University of California, San Francisco, CA 94143, USA. ✉email: bian.li@vanderbilt.edu; tony@capralab.org

The human proteome harbors millions of missense variants that could alter protein structure and function and contribute to disease risk[1]. Strong evolutionary constraint is a hallmark of sites critical to a protein's structure or function. A common approach to identifying constrained sites in human proteins has been to align the human protein sequence to those from other species and locate amino acid sites that are conserved across multiple species[2–5]. When combined with protein structures, this approach can facilitate generation of testable hypotheses about the structural mechanisms underlying the evolutionary constraint[6–9]. Such *interspecific* comparisons of sequences are powerful in detecting sequence conservation over long evolutionary timescales. Similarly, patterns of *intraspecific* coding variation in humans, especially low-frequency variants, also carry information about the functional importance of proteins and variants in human development and disease[1].

Leveraging ever growing human genetic variation data resources[1,10–12], several methods have been developed to estimate gene- or region-specific constraint based on tolerance to missense or loss-of-function variants in humans[1,12–18]. These gene-level and region-level measures of constraint have been effective in identifying Mendelian disease genes, genes under strong negative selection, and genes involved in severe neurodevelopmental disorders. However, some protein sites are critical for maintaining the integrity of protein structure or function, while others can be replaced with no or only minor impact on protein structure or function[19]. Metrics that yield a single score for an entire gene or a subregion do not capture the site-level variability in constraint that is essential for tasks such as interpreting the effects of specific variants of uncertain significance (VUS). To this end, the site-specific missense tolerance ratio (MTR), which compares the observed fraction of missense variation to the expectation under a null model within a sliding window of sites, was developed and shown to improve variant interpretation in epilepsy genes[20].

Recent analyses of the spatial distribution of missense variants in proteins have shown that population-level human standing variation can be analyzed in the context of 3D protein structures to identify specific regions and domains relevant to protein function and disease[21–26]. For example, analyses of tumor-derived somatic mutations within the context of protein structure indicate that variants tend to form spatial clusters and that these clusters often overlap functional domains in oncoproteins and tumor suppressors[23,27–30]. Analysis of 3D spatial patterns of both human germline and somatic variation also highlighted significant differences in the spatial mutational constraint on different classes of mutations in protein structure[21]. Recently, amino acid residue sites that are intolerant to missense variation have been characterized by incorporating protein structures and human genetic variation from large sequencing cohorts[22,25,26,31]. These studies suggest that missense variant analysis at the 3D level can identify functional sites and aid in variant interpretation. However, these previous analyses are limited by the availability of high-quality protein 3D structures and generally covered less than half of the proteins in the human reference proteome. In addition, while it is well-recognized that the mutability of individual amino acid sites is influenced by nucleotide sequence context[1,12,14,32–35] and that inter-residue spatial interactions are essential to maintaining structural and functional integrity of proteins[19], the consideration of the mutation spectrum at the resolution of native 3D interactions remains largely unexplored.

We hypothesize that connected functional sets of 3D neighboring amino acid sites, "contact sets", collectively shape the level of constraint on each site (e.g., as quantified by the depletion of missense variation compared to the amount expected under neutral evolution). We introduce the COntact Set MISsense tolerance (COSMIS) framework, to quantify the level of observed vs.

expected missense variation in a 3D structural context while correcting for nucleotide sequence context-dependent mutability of amino acid sites. We applied the framework to analyze the 3D spatial distribution patterns of 4.1 million unique missense variants at 6.1 million amino acid sites in their 3D structural context. We integrated high-quality protein 3D structures from three large resources, i.e., the Protein Data Bank (PDB)[36], the SWISS-MODEL repository[37], and the recently released, comprehensive database of protein 3D structures predicted by the AlphaFold2 algorithm[38,39]. Collectively, our framework covers 16,533 (80.3%) of all proteins in the human reference proteome. We show that our framework captures broad missense variant intolerance at the 3D spatial level across the human proteome. We demonstrate the utility of COSMIS in variant interpretation and in revealing structural insights into the pathogenic mechanisms of disease-causing variants. We further demonstrate the flexibility of the framework to work with custom-built homology models of potassium channels and with proteins in their oligomeric states. We propose that our COSMIS framework will have broad applicability in answering diverse questions about variant effect and to discover new genotype-phenotype relationships.

## Results

**The COSMIS framework maps 3D mutational constraint on proteins in high resolution.** We developed the COSMIS framework to quantify the 3D mutational constraint at each site in a protein structure by analyzing the patterns of genetic variants from large-scale sequencing projects in the context of protein structures (Fig. 1 and Methods). Our framework estimates the constraint on a site of interest (index site) as the depletion of missense variants in its 3D spatial neighborhood (i.e., contact set, see definition below) compared to the number expected if sites were evolving neutrally. We quantify this as the deviation of observed count of missense variants ($m_o$) from the expected count ($m_e$, accounting for transcript and codon missense mutability), divided by the standard deviation of the expected distribution ($m_\sigma$). We designate this Z score as the COSMIS score and assign it to the index site. We obtain the observed variant count in each contact set by mapping variants cataloged by the Genome Aggregation Database (gnomAD, v2.1.1)[1] onto protein structures. To obtain the expected variant count distribution, we use a procedure that simulates mutation under neutral evolution based on a 64 by 3 sequence-context-dependent mutability matrix derived from the 1000 Genomes Project variant set[11]. We also compute an empirical $p$ value for each COSMIS score based on the simulation procedure and the resulting expected count distribution (Methods). According to our formulation of the score, a lower value indicates a greater depletion of missense variants in the spatial neighborhood and hence lower missense variation tolerance.

**3D structural context differs from 1D sequence context.** COSMIS scores are based on the 3D interaction context of protein-coding sites. We quantify this context using contact sets, defined as the set of amino acid residues that are in contact with the residue located at the index site (Fig. 2a). A pair of residues are considered to be in contact when the distance between their $C_\beta$ atoms (or $C_\alpha$ atoms in the case of glycines) is less than 8 Å, a threshold commonly used to define residue contact[40]. To compute contact sets, we collected high-quality protein 3D structures (Methods and Supplementary Fig. 1) that collectively cover 80.3% of all proteins in the human reference proteome (UP000005640, UniProt release 2021_03) from the PDB[36], SWISS-MODEL repository[37], or the AlphaFold database of highly accurate predicted structures (AF2)[38,39] (Fig. 2b) (Supplementary Data 1).

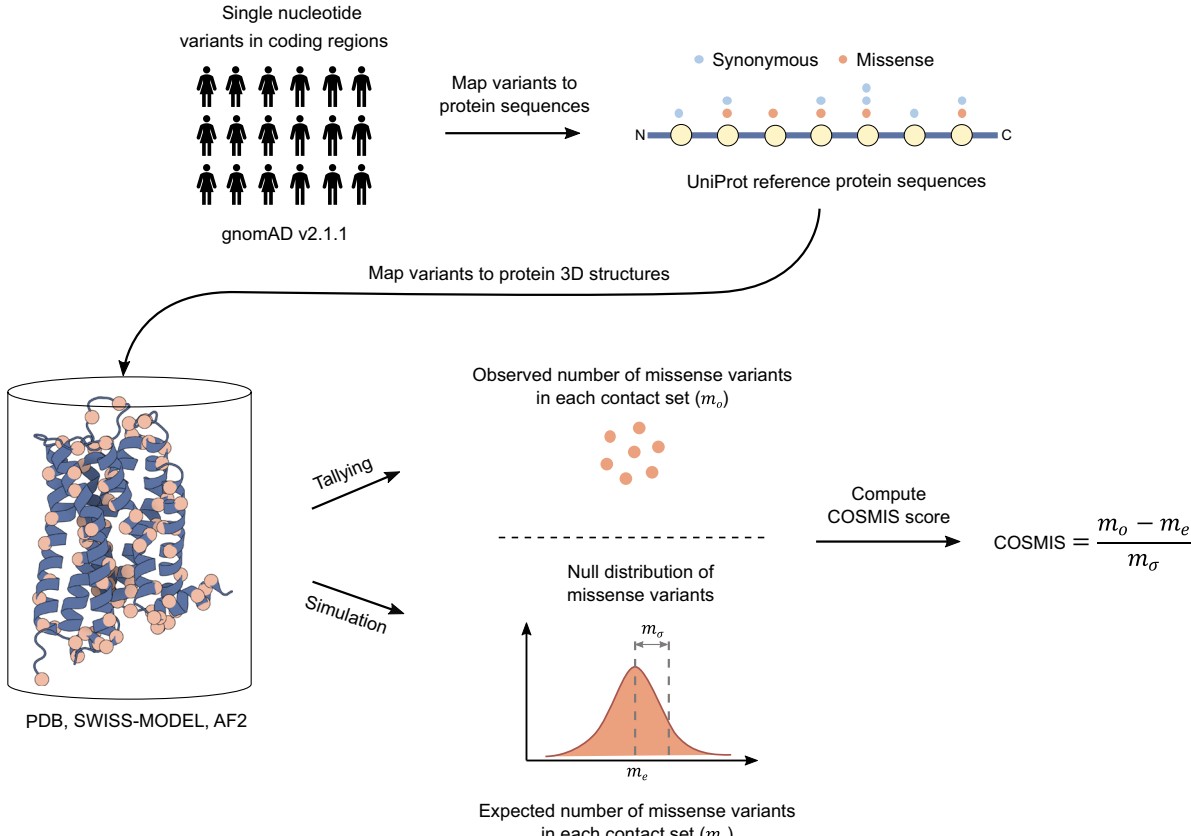

**Fig. 1 Schematic of the COSMIS 3D mutational constraint quantification framework.** The COSMIS framework consists of mapping single nucleotide variants (SNVs) from gnomAD to human reference protein sequences and protein 3D structures, the computation of 3D contact sets, tallying of unique missense variants observed in contact sets, and comparison of observed missense variant counts to a null distribution simulated based on a mutation-spectrum-aware statistical model. We quantify the constraint on the index amino acid site with the Z score of the observed missense variant count ($m_o$) compared to the null distribution. gnomAD Genome Aggregation Database, PDB Protein Data Bank, AF2 AlphaFold2.

Our framework also has high residue-level coverage. Structures from PDB, SWISS-MODEL, and AF2 have a median residue-level coverage of 78.0%, 70.9%, and 76.7%, respectively (Supplementary Fig. 1). Collectively, we computed contact sets for each of 6.1 million unique index sites in the human reference proteome (Fig. 2b). For simplicity, we report results for experimental structures and computational models together, since patterns were similar when we analyzed them separately (Supplementary Figs. 2 and 3).

Conceptually, a contact set captures residues that are close in 3D space, even when they are far apart in sequence. Our analysis shows that contact sets include critical "long-range" (defined here as >15 residues apart along 1D sequence) residue-residue interactions that would be missed by 1D sequence-based metrics that only consider a window around the index site. For example, for 17.9% of all 6.1 million amino acid sites, at least 50% of the 3D contacts they make are long-range, and 66.3% of all sites make at least 10% long-range 3D contacts (Fig. 2c). On the other hand, windows based on sequence context alone contain many sites that are not in 3D contact with the index site. For example, all of the 6.1 million sites in this study have at least 50% of their 30 1D sequence neighbors (15 sites on each side) not in 3D contact, and nearly half (47.5%) of all sites have least 80% of sequence neighbors not in 3D contact (Fig. 2d). Thus, long-range 3D contacts are common, as are sites that are nearby in sequence but distant in 3D. (Additional statistics about long-range 3D contacts at 6 Å and 10 Å distance thresholds are available in Supplementary Fig. 4.) Residues in 3D contact are likely to be essential for the structural stability and functional integrity of the residue at

the index site; thus, 3D structure-based residue contact sets give a more sensitive representation of the structural and function context of coding sites than sequence-based windows.

**COSMIS captures constraint at both protein and site levels**. Consistent with our expectation and previous observations at the gene level, our framework identifies broad constraint on missense variants at the protein level and little constraint on synonymous variants[1],[12]. We computed the deviation from the expected count (observed - expected) at the protein level for both synonymous and missense variants across the entire dataset. Supporting our approach for estimating the expected variant count distributions (Methods), the deviation between the observed and expected synonymous variant count is low and centered near zero (Fig. 3a; median 1.9, standard deviation 39.4). In contrast, the difference between observed and expected is significantly shifted toward negative values for missense variants (Fig. 3a; median −29.1, standard deviation 104.8, $p < 2.2 \times 10^{-308}$, two-sided Mann–Whitney $U$ test).

We next computed COSMIS scores for the 6.1 million unique amino acid sites across 16,533 proteins in the human reference proteome with sufficient data (Fig. 3b and Supplementary Fig. 5). As expected, the distribution of scores spans a wide range from negative (constrained) to positive (unconstrained) values, with a significant shift toward constraint (median −0.47, standard deviation 1.2). Proteins with experimentally determined structures in the PDB have a significantly lower median COSMIS score than those currently only have computationally predicted structures in SWISS-MODEL and AF2 databases (median −0.62 vs. −0.42 and

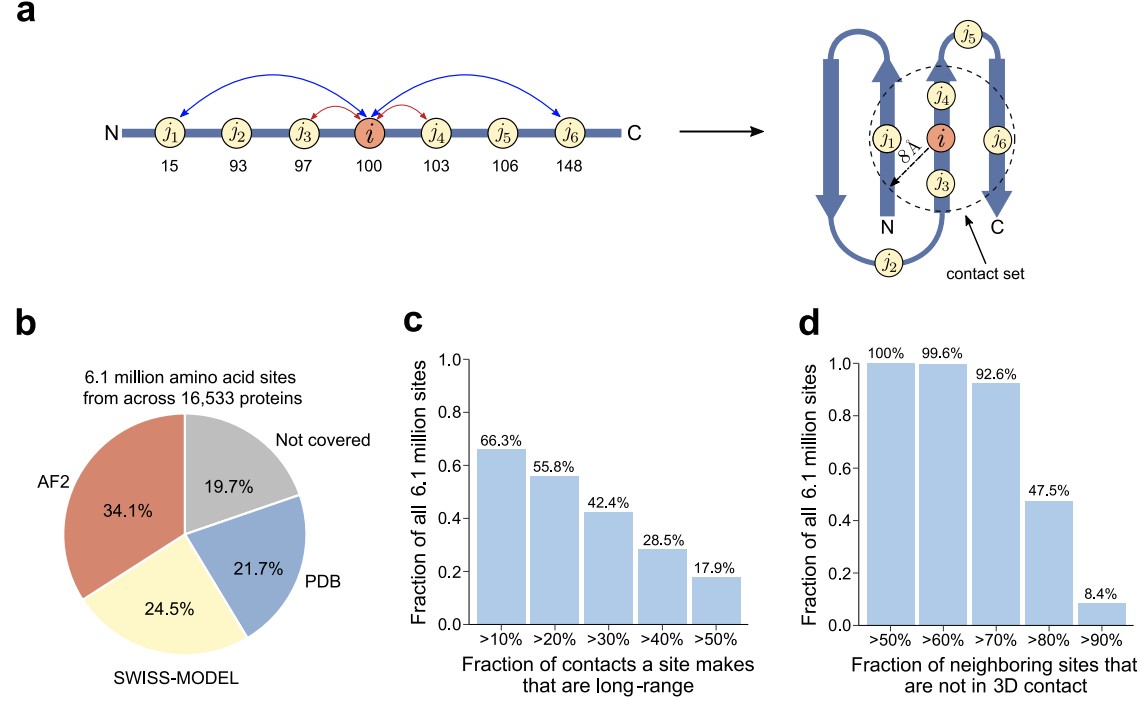

**Fig. 2 Protein 3D context differs from 1D sequence context. a** To quantify the 3D spatial context of each amino acid site (*i*), our framework defines its contact set as the amino acid residues that are in contact ($C_\beta < 8$ Å) with the residue. For the example index site (*i*), the contact set is (*i*, $j_1$, $j_3$, $j_4$, $j_6$). Numbers below the 1D sequence schematic represent residue sequence positions and illustrate that contact set residues may be distant in sequence from the index site. **b** The COSMIS framework covers 80.3% of the reference human proteome. Defining the contact set of an amino acid site requires protein 3D structures. We used PDB and SWISS-MODEL as our primary sources of protein 3D structures. For proteins with no structure in the PDB or SWISS-MODEL that meet our criteria (Methods), we analyze models from the AlphaFold2 structure database. Numbers inside the pie chart represent fractions of the human reference proteome (20,600 proteins) for which we used the corresponding protein structure resource to compute COSMIS scores (Supplementary Data 1). **c** Contact sets capture long-range sites (separated by more than 15 residues along the 1D sequence) that interact in 3D. For example, residues $j_1$ and $j_6$ in panel **a** are not neighbors in 1D sequence, but nevertheless form long-range contacts with the index site *i*. The bar plot shows the fraction of all 6.1 million sites with at least a certain fraction of long-range 3D contacts in their contact sets. **d** Many neighboring sites in 1D sequence do not form 3D contacts with an index site. Defining the contact set eliminates these sites from consideration. For example, residues $j_2$ and $j_4$ in panel **a** are 1D sequence neighbors (within 15 residues) of the index site *i* but do not form 3D contacts with it. The bar plot shows the fraction of all 6.1 million sites that have at least a certain fraction of 1D sequence neighbors that do not form 3D contacts. PDB Protein Data Bank, AF2 AlphaFold2. Source data are provided as a Source Data file.

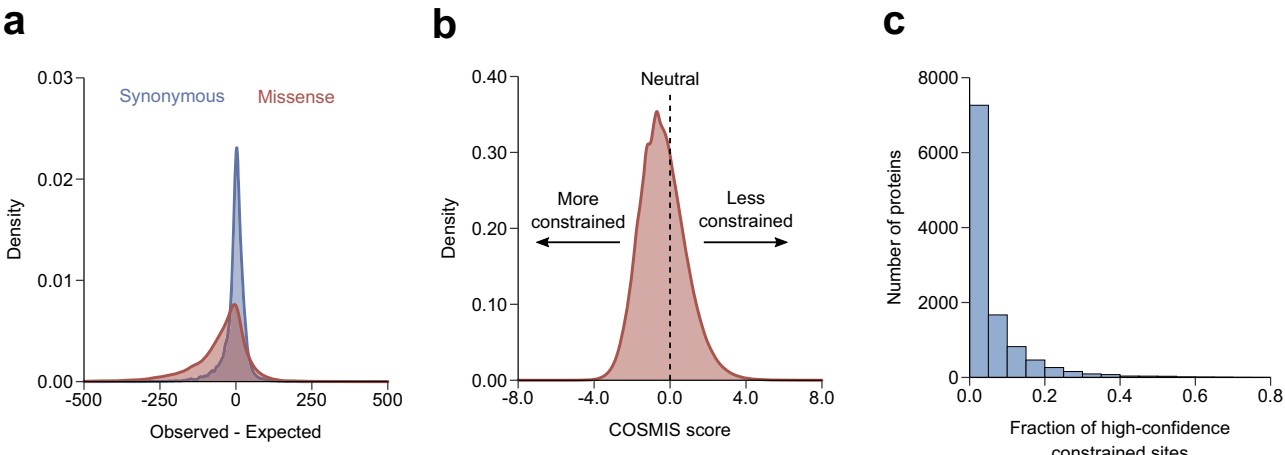

**Fig. 3 The COSMIS score quantifies depletion of missense variants in contact sets. a** Distribution of the deviation of the observed from the expected number of synonymous (blue) and missense (red) variants per-transcript computed from the mutability-aware model (Methods). The median of the deviation is roughly centered at zero (1.9) for synonymous variants but is significantly shifted towards negative values (more constraint) for missense variants (median $-29.1$, $p < 2.2 \times 10^{-308}$, two-sided Mann–Whitney $U$ test). **b** Distribution of the COSMIS scores for 6.1 million unique amino acid sites of the reference human proteome. As expected, an average amino acid site in the human proteome is depleted of missense variants in its contact set (median COSMIS score $-0.47$) due to structural and/or functional constraint. **c** Distribution of per-protein fraction of high-confidence constrained sites (empirical $p < 0.01$). Source data are provided as a Source Data file.

0.44, respectively, $p < 2.2 \times 10^{-308}$, two-sided Mann–Whitney $U$ test) (Supplementary Fig. 6), suggesting that proteins with greater functional importance have historically been selected for structural characterization. In contrast, a similarly constructed score based on synonymous variants (i.e., contact set synonymous tolerance score) is centered at 0, regardless of the sources of protein 3D structures (Supplementary Fig. 6), consistent with overall results and the hypothesis that synonymous variants are not subject to 3D mutational constraint in protein structures.

We consider sites with a COSMIS score for which the empirical $p$ value obtained from simulation is <0.01 as high-confidence (this is approximately equivalent to COSMIS score < −2.33). Overall, we find 313,204 sites (5.1%) with high-confidence constraint scores from 10,955 proteins (66.3%), with an average of 28.6 high-confidence constrained sites per protein (Fig. 3c, Supplementary Data 2). These sites are generally clustered in 3D space. The average pairwise distance between high-confidence constrained sites is on median only 66% of the distance expected from random permutations (Supplementary Fig. 7), suggesting that high-confidence constrained sites identified by COSMIS likely represent functionally important domains. Overall, these findings suggest that the COSMIS score captures the depletion of missense variants in 3D structure-based contact sets resulting from varying functional constraint over protein space.

**COSMIS refines gene-level constraint estimates.** With the growth of large human genetic variation datasets, methods have been developed to quantify constraint on individual sites, sequence windows, and genes. Given the connection between constraint and function, constraint scores are critical components of many gene and variant interpretation tasks. For example, gene-level metrics, like pLI[12], have been extensively used to prioritize genes in which variants are likely to contribute disease risk. However, gene-level metrics do not identify amino acid sites within each gene that are under constraint. pLI accurately identifies gene-level constraint, but frequently classifies genes that harbor known disease-associated mutations as loss-of-function (LoF) tolerant[41]. This is not surprising, and illustrates a weakness of gene- and region-level metrics. Site- and window-based intraspecies constraint metrics provide a higher resolution view, but as demonstrated above, sequence context is often very different from 3D structural interaction context. Since COSMIS quantifies constraint at the contact set level for each amino acid site, we hypothesized that it would provide a higher-resolution view of the clinical importance of protein regions, in addition to capturing broad constraint at the protein level.

To explore this hypothesis, we computed the distribution of per-protein COSMIS scores for 16,260 proteins stratified into the three pLI classes (Intolerant ($n = 2566$), Unsure ($n = 2900$), Tolerant ($n = 10794$)). (We were not able to obtain the pLI scores for 273 proteins with COSMIS scores.) As expected, on average LoF intolerant genes (pLI ≥ 0.9) have significantly lower COSMIS scores than LoF tolerant genes (pLI ≤ 0.1) (−1.1 vs. −0.12, $p < 2.2 \times 10^{-308}$, two-sided Mann–Whitney $U$ test, Fig. 4a), indicating that sites in LoF intolerant genes are on average more constrained than LoF tolerant genes. Genes that have medium pLI scores (0.1 < pLI < 0.9) also have medium COSMIS score on average (−0.80).

While LoF tolerant genes have less evidence of 3D mutational constraint overall, we found that 1888 (40.4%) LoF tolerant genes have at least one high confidence constrained site (COSMIS score < −2.33), with 13.6 on average. For example, the ubiquitin-like modifier-activating enzyme 5 (UBA5) is considered LoF tolerant (pLI score of $2.5 \times 10^{-4}$). However, our analysis indicates that UBA5 has many constrained sites in interfaces of UBA5

dimerization, UBA5-UFM1 binding, and UBA5-ATP interaction (Fig. 4b, c). Specifically, of the 30 (10%) most constrained sites in UBA5, 13 sites are located at the UBA5 dimerization interface, four sites interact with ATP, and another three are involved in UFM1 binding (Fig. 4c). This is consistent with UBA5's involvement in severe epileptic encephalopathy[42]. Indeed, the three constrained sites with the strongest constraint according to COSMIS (amino acid residues 54, 57, and 58) include M57V, which was found in a patient cohort to drastically reduce UBA5's catalytic activity[42]. Thus, the COSMIS scores of UBA5 identify amino acid sites relevant to UBA5's functions and known disease associations. This illustrates how considering constraint in spatial neighborhoods can identify genes predicted to be LoF tolerant (low pLI) that are clinically important and suggests that COSMIS can guide further investigation before discarding genes from clinical consideration.

**COSMIS highlights pathogenic variants and essential proteins.** To quantify the ability of COSMIS to contribute to identification of disease-associated protein variants, we compared the COSMIS scores for a total of 19,346 unique sites harboring benign and 14,824 unique sites harboring pathogenic missense variants with unambiguous annotations of clinical significance and 115,172 VUS sites from ClinVar (Methods, Supplementary Data 3). Pathogenic variants have a significantly lower COSMIS score distribution than benign variants (median −1.1 vs. 0.0, respectively; $p < 2.2 \times 10^{-308}$, two-sided Mann–Whitney $U$ test; Fig. 5a). The median COSMIS score of the VUS set is −0.31, consistent with the expectation that it is a mixture of variants of various functional effects. The distance threshold for defining residue contact has little effect on the score distributions of these variant sets relative to each other (Supplementary Fig. 8). Further division of variants into four subgroups, i.e., benign, likely benign, likely pathogenic, and pathogenic, shows that the median score of likely benign variants is slightly lower than that of benign variants (−0.12 vs. 0.07; $p = 3.3 \times 10^{-23}$, two-sided Mann–Whitney $U$ test), whereas pathogenic and likely pathogenic variants both have lower scores (median −1.12 vs. −1.17, respectively; $p = 0.01$, two-sided Mann–Whitney $U$ test) (Supplementary Fig. 9). Collectively, the significant negative shift for pathogenic variants suggests strong constraint in their spatial neighborhoods, while the average neutral COSMIS score of benign variants suggests less constraint on missense variants in their contact sets.

Across the COSMIS score range, the magnitude of the score correlates with enrichment for pathogenic over benign variants. High-confidence constrained sites are 13.5-fold enriched for pathogenic variants. However, only 1706 out of the 10,955 proteins that have at least one high-confidence site have unambiguously annotated pathogenic variants in ClinVar (Supplementary Data 4), suggesting that many pathogenic variants are yet to be uncovered. Moving down the constraint spectrum, the top 10% most constrained COSMIS sites (equivalent to COSMIS score < −1.85) are 10.6-fold enriched for pathogenic variants, and the bottom 10% are 3.3-fold depleted (Fig. 5b). Our analysis suggests that constraint on missense variation in a site's 3D interaction context (as quantified by COSMIS) is strongly correlated with variant pathogenicity.

To evaluate the relationship between 3D mutational constraint as quantified by COSMIS and function and disease associations at the protein level, we compared the COSMIS score distributions of amino acid sites in six groups of proteins encoded by genes expected to be under various levels of constraint (and the dataset as a whole). In general, proteins with essential functions and disease associations have lower COSMIS scores than proteins without (Fig. 5c) and as the essentiality of a gene increases, amino

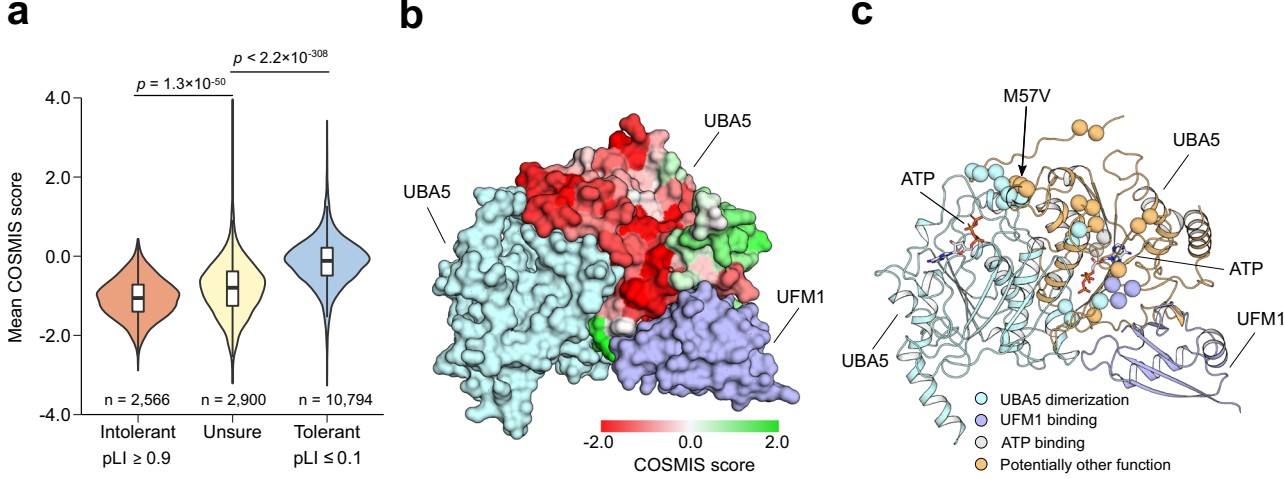

**Fig. 4 The COSMIS score quantifies constraint at amino acid resolution and provides structural insights into variant pathogenicity. a** COSMIS score distributions are significantly different between loss of function tolerant, unsure, and intolerant (as defined by pLI) genes. The COSMIS scores of amino acid sites in intolerant genes differ from those in tolerant genes (median −1.1 vs −0.12). Statistic test: two-sided Mann–Whitney $U$ test. In boxplot graphs center line indicates median, bounds of box indicate 25th and 75th percentiles, and whiskers indicate minimum and maximum. **b** COSMIS scores of UBA5 sites mapped to structure of one subunit of a dimerized UBA5 bound with the UFM1 target protein (PDB ID: 6H77). UBA5 is predicted to be LoF tolerant, but it exhibits substantial constraint on specific spatial regions. Structures of all subunits of the complex are rendered in surface. **c** Locations of the top 10% most constrained sites in UBA5 ranked by COSMIS score. Sites are rendered in spheres and colored according to their likely functional roles. Location of variant M57V implicated in early-onset encephalopathy is indicated. Proteins are rendered in cartoons. We note that because the COSMIS score of a site is directly informed by the genetic variability of its contact set, it comes as natural to interpret the scores in the context of a 3D structure. pLI probability of loss-of-function intolerant, UBA5 ubiquitin-like modifier-activating enzyme 5, UFM1 ubiquitin-fold modifier 1, ATP adenosine triphosphate. Source data are provided as a Source Data file.

acid sites in the encoded protein have more negative COSMIS scores on average. Haploinsufficient genes (a single-copy of the functional allele is insufficient to produce the expected phenotype)[43], genes essential in cell culture, and genes associated with dominant diseases encode proteins that have the lowest COSMIS score distributions among all evaluated categories. In contrast, constrained sites are much less frequently found in proteins encoded by nonessential genes. Not surprisingly, olfactory receptors[44] have the least 3D mutational constraint of any protein groups considered. The abundance of high-confidence constrained sites in each protein follows the same general trend (Supplementary Fig. 10). Our analysis identifies 72 proteins with more than 50% high-confidence constrained sites (Supplementary Data 5). These proteins are likely to be under extreme purifying selection. In fact, it has been suggested that 10 of these proteins are either essential[45] or harbor variants that are haploinsufficient[43] or associated with diseases that follow dominant inheritance[46,47] (Supplementary Data 5). Overall, our analysis indicates that the COSMIS score strongly reflects functional constraint and is predictive of variant pathogenicity.

**COSMIS complements existing quantifications of intra- and inter-species constraint.** To assess the relationship between COSMIS and other intra- and interspecies constraint metrics, we first compared COSMIS to four commonly used intraspecies constraint metrics that do not consider structural context (MTR, RVIS, pLI, and Missense_Z). We compared these other metrics to COSMIS in their ability to identify pathogenic variants using a total of 8063 benign and 7257 pathogenic missense variants from ClinVar for which all scores could be computed (Supplementary Data 6). COSMIS achieved a significantly higher area under the receiver operating characteristic curve (AUROC) than the other intraspecies constraint metrics (e.g., 0.733 vs. 0.653 for COSMIS vs. MTR, $p = 1.0 \times 10^{-65}$, two-sided DeLong's test, Fig. 6a). To illustrate these patterns, analysis of the COSMIS scores for a set of

functionally characterized VUS in the *SCN5A* sodium channel[48] shows that COSMIS "rescues" pathogenic variants that would be misclassified by MTR (Supplementary Fig. 11; Supplementary Data 7). These results suggest that 3D neighboring residues contribute critical information about the functional importance of index sites. We additionally compared COSMIS to a recently developed version of MTR that considers missense variants in 3D neighborhoods (MTR3D), but does not account for sequence context-dependent mutability[25,26]. COSMIS also performs significantly better than MTR3D (i.e., 0.733 vs. 0.665, $p = 2.5 \times 10^{-50}$, two-sided DeLong's test, Fig. 6a), suggesting that accounting for the variability of mutability is essential to estimating constraint.

We then compiled a subset of 3.6 million amino acid sites for which the five commonly used intraspecies constraint metrics (MTR, MTR3D, RVIS, pLI, and Missense_Z) could also be computed. To summarize the relationships between these constraint scores, we computed their pairwise Spearman correlations across sites (Fig. 6b). As expected, pLI and Missense_Z have the highest Spearman's ρ (0.63), given that they both quantify gene-level constraint and were derived with similar approaches[12]. Similarly, MTR and MTR3D are well correlated (Spearman's ρ 0.53). The COSMIS score has a comparable level of correlation with both MTR and MTR3D (0.41 and 0.39). The intermediate correlations suggest that the metrics capture constraint at different scales, as expected.

To illustrate the differences between the *intra*species constraint scores and *inter*species phylogenetic conservation metrics, we additionally computed the correlations for four common interspecies phylogenetic conservation metrics (GERP++, phyloP, phastCons, ConSurf). Phylogenetic conservation metrics are generally more correlated with each other than with any of the intraspecies constraint scores (Fig. 6b). For example, the lowest Spearman's ρ between any pair of the four phylogenetic conservation metrics is 0.47 (GERP++ vs. phyloP), higher than

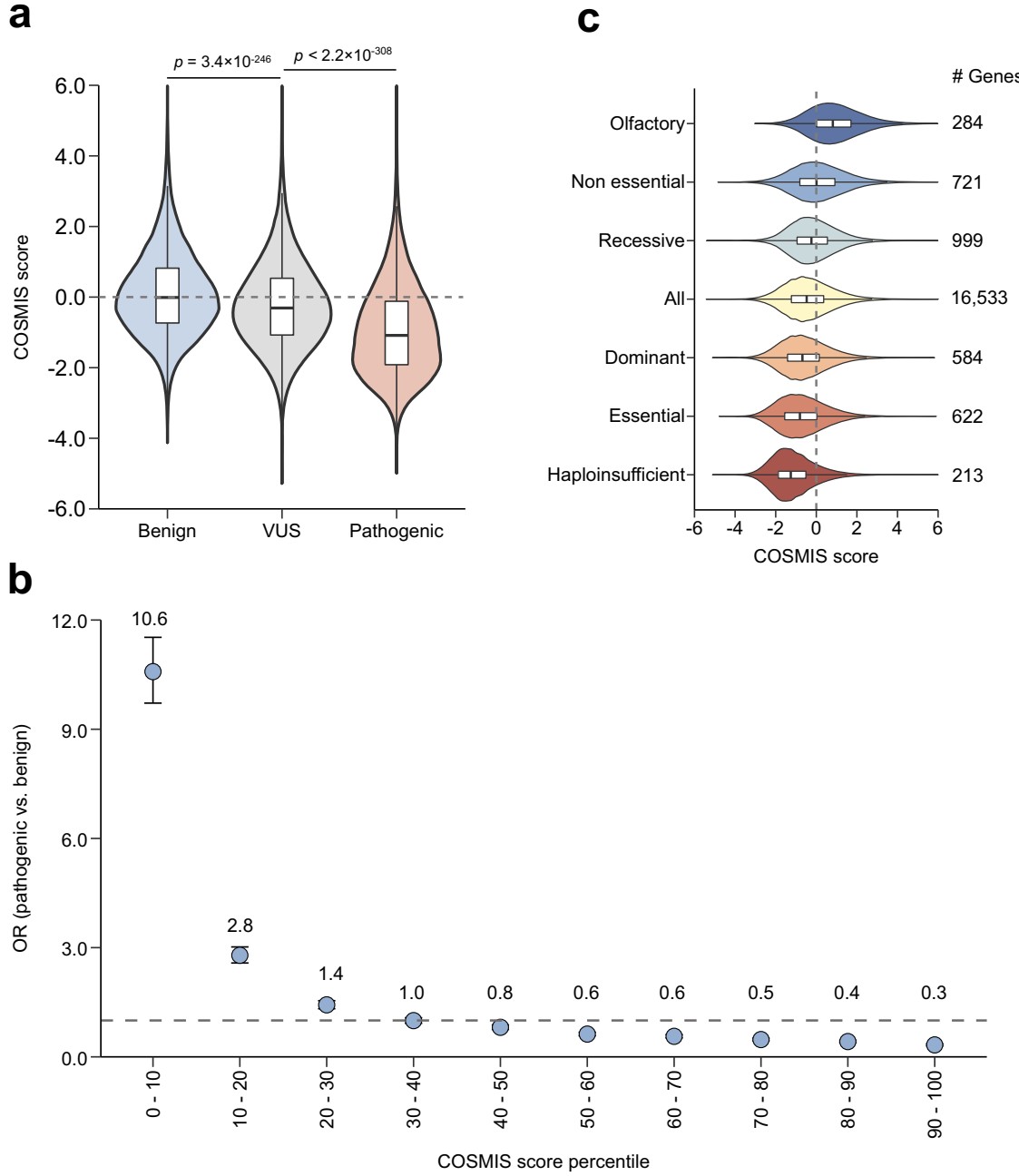

**Fig. 5 The COSMIS score is strongly correlated with both pathogenicity and gene constraint level. a** COSMIS score distributions for 19,346 benign, 14,824 pathogenic, and 115,172 VUS sites from ClinVar (Methods). Pathogenic variants have significantly more constrained 3D spatial neighborhoods (COSMIS score median −1.1) than benign variants (median score 0.0) ($p < 2.2 \times 10^{-308}$, two-sided Mann–Whitney $U$ test). In boxplot graphs center line indicates median, bounds of box indicate 25th and 75th percentiles, and whiskers indicate minimum and maximum. **b** Odds ratio of ClinVar pathogenic variants versus benign variants for different COSMIS score percentile bins (lower bins correspond to more constrained COSMIS scores). Amino acid sites with lower COSMIS scores are enriched for pathogenic variants whereas sites with higher scores are depleted of pathogenic variants. Error bars indicate 95% confidence intervals. The horizontal dashed line indicates OR = 1. The values for each cell of the contingency table used for the OR calculation in each percentile bin were reported in Supplementary Data 14. **c** COSMIS score distributions of amino acid sites in six groups of proteins encoded by genes with different functional annotations (and the dataset as a whole). As the anticipated functional constraint on each category increases (top-to-bottom), amino acid sites in proteins in the category have more constrained COSMIS scores on average. In boxplot graphs center line indicates median, bounds of box indicate 25th and 75th percentiles, and whiskers indicate minimum and maximum. OR odds ratio, VUS variants of uncertain significance. Source data are provided as a Source Data file.

the highest Spearman's ρ between a phylogenetic metric and an intraspecies constraint score (i.e., 0.38, ConSurf vs. RVIS and Missense_Z). This is consistent with previous finding that intraspecies constraint metrics are only modestly correlated with phylogenetic conservation[16,17]. Together with the observation

that both groups of metrics demonstrated predictive ability for variant pathogenicity (Fig. 6a and Supplementary Fig. 12), this suggests that these two groups of metrics contain complementary information that, when combined, could improve performance in predicting variant pathogenicity.

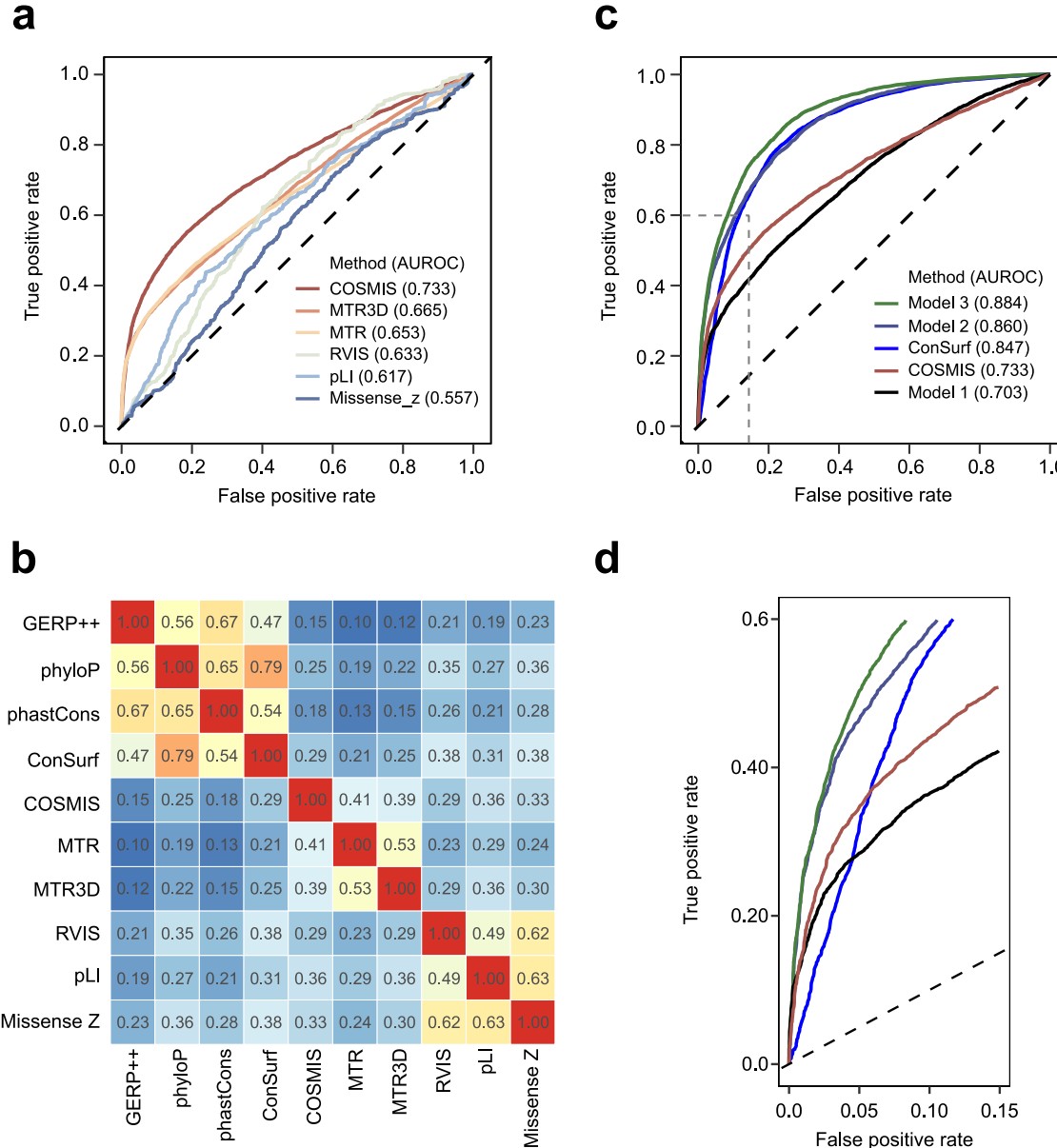

**Fig. 6 The COSMIS score is more predictive of variant pathogenicity than other constraint metrics. a** Comparison of the performance of COSMIS with five other constraint scores in predicting the pathogenicity of a total of 8063 benign and 7257 pathogenic missense variants from ClinVar for which all scores are available. COSMIS significantly outperforms the other methods (AUROC 0.733 vs. 0.665 for MTR3D, the next best-performing method, $p = 2.5 \times 10^{-50}$, two-sided DeLong's test). **b** A heatmap of Spearman rank correlations (absolute values) between four phylogenetic conservation scores (GERP++, phyloP, phastCons, ConSurf) and six constraint scores (COSMIS, MTR, MTR3D, RVIS, pLI, Missense Z) that are constructed based on human population genetic variants. **c** COSMIS is complementary to phylogenetic constraint methods. ROC curves of logistic regression models integrating different combinations of the ten methods in panel **b** at predicting the pathogenicity of the variants from ClinVar. Model 1: MTR + MTR3D + RVIS + pLI + Missense Z; Model 2: COSMIS + ConSurf; Model 3: all scores + relative solvent accessibility. **d** A zoomed-in view of the high-confidence region of ROC space (bounded by the dashed lines in **c**. The improvement from adding COSMIS to ConSurf over ConSurf alone is mainly due to better performance in this high-confidence region. AUROC area under the receiver operating characteristic. Source data are provided as a Source Data file.

Given this, we hypothesized that integrating interspecies scores with COSMIS could provide additional information for pathogenicity prediction. To test this hypothesis, we combined COSMIS with ConSurf (the best-performing interspecies metric on our dataset, Supplementary Fig. 12) using a logistic regression model and evaluated the resulting performances with five-fold cross validation (Methods). Our evaluation shows that integrating COSMIS and ConSurf outperformed the AUROC of both ConSurf and COSMIS alone (0.860 vs. 0.847 and 0.733, $p = 0.002$ and $p = 6.3 \times 10^{-145}$, respectively, two-sided DeLong's test, Fig. 6c). In particular, the improvement from adding

COSMIS to ConSurf over ConSurf alone is mainly due to better performance in the high-confidence region (Fig. 6d). However, we note that ConSurf alone outperforms COSMIS alone (0.847 vs. 0.733) (Fig. 6c). Combining all 10 scores and relative solvent accessibility in a logistic regression model resulted in additional AUROC improvement (0.884 vs. 0.860, $p = 6.0 \times 10^{-10}$, two-sided DeLong's test, Fig. 6c). Our results suggest that COSMIS score contributes additional information to phylogenetic conservation for pathogenicity prediction and that adding intraspecies constraint can improve the performance of even the best phylogenetic conservation scores.

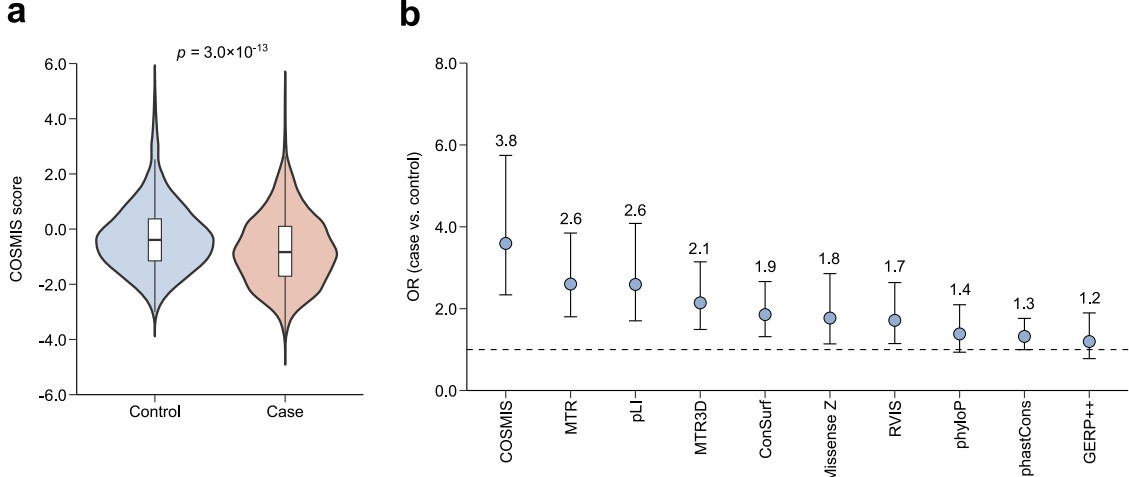

**Fig. 7 COSMIS score improves interpretation of de novo missense mutations from neurodevelopmental disorders. a** COSMIS score distributions for de novo missense mutations from neurodevelopmental disorder cases (Case) and from unaffected siblings of autism spectrum disorder probands (Control). Case variants ($n = 2271$) have a significantly more constrained spatial neighborhoods than control variants ($n = 571$) (median COSMIS $-0.83$ vs. $-0.39$, $p = 3.0 \times 10^{-13}$, two-sided Mann–Whitney $U$ test). **b** Case variant enrichment analysis for intra- and inter-species constraint metrics at the $10^{th}$ percentile of most constrained sites. COSMIS has the highest enrichment for cases (OR 3.6, 95% confidence interval [2.3, 5.7]). Error bars are 95% confidence intervals of ORs. Results of this OR analysis are consistent across thresholds other than the $10^{th}$ percentile (Supplementary Fig. 9). The values for each cell of the contingency table used for the OR calculation in each percentile bin were reported in Supplementary Data 15. OR odds ratio. Source data are provided as a Source Data file.

**COSMIS distinguishes de novo variants in neurodevelopmental disease cases and controls**. De novo variants are often clinically relevant and are more likely to be pathogenic than inherited variation[49]; however, they are difficult to interpret. De novo variants play a prominent role in rare and common forms of neurodevelopmental disorders[50], and de novo variants in neurodevelopmental disease cohorts have been used previously to benchmark the utility of constraint metrics for variant interpretation[14]. To test if considering 3D mutational constraint could contribute to the interpretation of de novo variants, we compared the COSMIS distributions for 2271 de novo missense variants from neurodevelopmental disorder probands (case variants) versus 541 de novo missense variants from unaffected siblings of autism spectrum disorder probands[51] for which COSMIS scores can be computed (Supplementary Data 8). Control variants had a median COSMIS score significantly higher than the median COSMIS score of case variants ($-0.39$ vs. $-0.83$, $p = 3.0 \times 10^{-13}$, two-sided Mann–Whitney $U$ test) (Fig. 7a).

For context, we compared the ability of COSMIS to enrich for case variants with the other inter- and intraspecies metrics considered previously. We did this analysis using 1,506 case and 306 control variants for which all 10 scores are available (Supplementary Data 9). For COSMIS, 24.2% case and 8.2% control variants fall within the $10^{th}$ percentile of most constrained sites (i.e., COSMIS score $< -1.85$), corresponding to an odds ratio (OR) of 3.6 ($p = 2.6 \times 10^{-11}$, two-sided Fisher's exact test) (Fig. 7b). Both MTR and pLI achieved the next highest OR of 2.6 at the $10^{th}$ percentile of most constrained sites, while being lower than COSMIS. Except for MTR3D, which has an OR of 2.1, the ORs of all other metrics are below 2 (Fig. 7b). COSMIS also has the highest ORs at other thresholds ($5^{th}$ and $20^{th}$ percentiles) (Supplementary Fig. 13). The modest performance of all evaluated metrics (including COSMIS) is not surprising as we do not expect all de novo variants in cases to be causal/pathogenic[52]. To more accurately benchmark the ability of these metrics to predict the pathogenicity of specific de novo variants, a well-established set of de novo variants with clinically validated disease associations is needed. Nevertheless, our analysis indicates

that reliably prioritizing de novo variants for further investigation is still a challenging problem for contemporary metrics, as has been previously suggested[17,53].

**Applying COSMIS to custom-built oligomeric potassium channel structures**. While we have precomputed COSMIS scores for >80% human proteins using publicly available structures, it is conceivable that the structure of a protein of interest might not be available in public databases. To demonstrate the flexibility of our framework to work with protein 3D structures built using macromolecular modeling tools and to investigate whether COSMIS score could capture constraint imposed by protein-protein oligomerization[54–56], we compiled a set of 41 potassium ion channel (KCN) genes (Supplementary Data 10) for which variants have been annotated in ClinVar. KCN genes encode proteins that function in obligate homo-oligomeric states[57], so we expected interface sites to be under stronger constraint than non-interface sites. We obtained structures for these KCN proteins in their homo-oligomeric states either from the PDB or through homology modeling using the SWISS-MODEL interactive workspace[37]. Collectively, we structure-mapped and computed two sets of COSMIS scores, based on monomers and oligomers, respectively, for 4762 interface and 14,331 non-interface sites in these potassium channels. As expected, we found that on average interface sites are significantly more constrained than non-interface sites (median COSMIS score $-1.3$ vs. $-1.1$, $p = 1.3 \times 10^{-16}$, two-sided Mann–Whitney $U$ test; Fig. 8a). When computed based on oligomer structures, the scores of interface sites shift significantly to more negative values (median difference $-0.13$, $p = 8.1 \times 10^{-8}$, two-sided Mann–Whitney $U$ test; Fig. 8b). This was further confirmed by analyzing the scores of a larger set of 1678 diverse human homodimers from the PDB (median difference $-0.11$, $p = 3.1 \times 10^{-12}$, two-sided Mann–Whitney $U$ test; Methods; Supplementary Fig. 14), suggesting that COSMIS captures additional constraint on interface sites contributed by sites in neighboring subunits.

We next evaluated the performance of COSMIS in predicting the pathogenicity of missense variants in KCN oligomers.

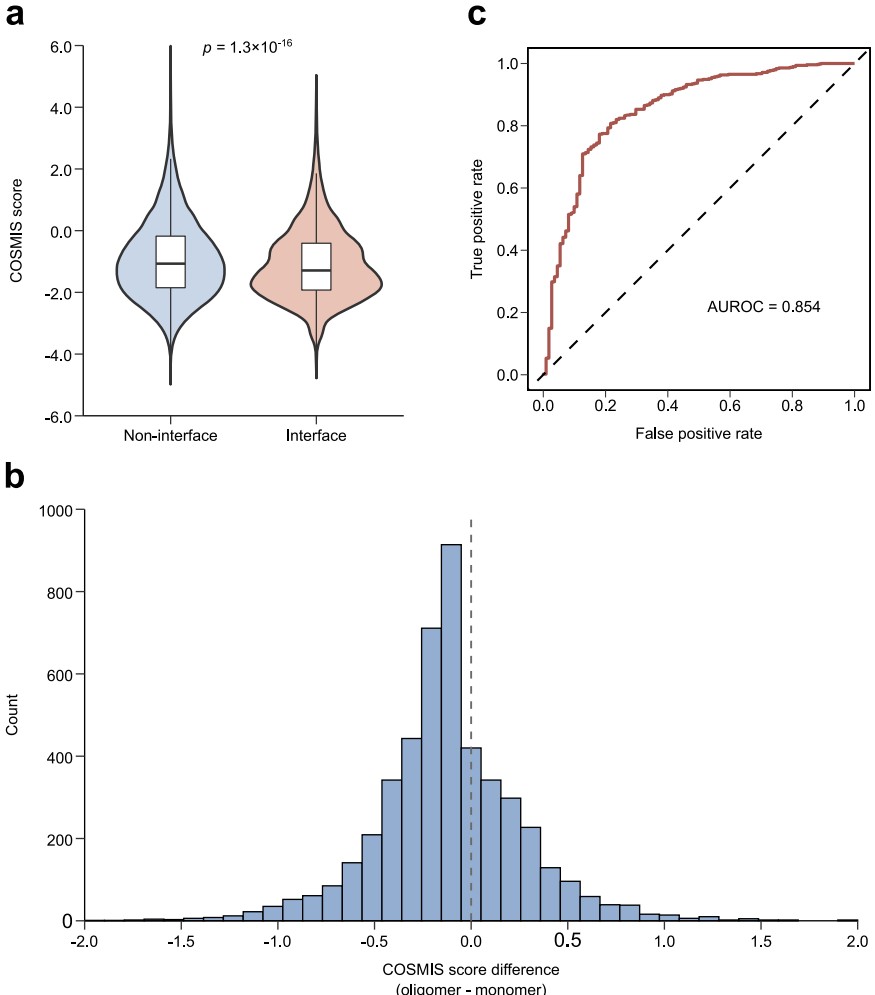

**Fig. 8 Applying COSMIS to custom-built oligomeric structural models facilitates interpretation of potassium channel variants. a** COSMIS score distributions for interface and non-interface amino acid sites in 41 oligomeric potassium ion channels. Overall, interface sites ($n = 4762$) involved in oligomerization (making more 3D contacts in oligomers than in monomers) have a significantly lower COSMIS scores than non-interface sites ($n = 14,331$) (median $-1.3$ vs. $-1.1$, $p = 1.3 \times 10^{-16}$, two-sided Mann–Whitney $U$ test). **b** COSMIS scores of interface sites computed based on oligomers are generally lower than those computed based on monomers (median difference $-0.13$, $p = 8.1 \times 10^{-8}$, two-sided Mann–Whitney $U$ test). **c** COSMIS score performs well (AUROC 0.854) at predicting the pathogenicity of 111 benign and 489 pathogenic potassium channel missense variants curated from ClinVar. AUROC area under the receiver operating characteristic. Source data are provided as a Source Data file.

Establishing the pathogenicity of variants in KCN genes is clinically significant, because they have been associated with multiple severe neurological, psychiatric, and cardiac disorders such as epileptic encephalopathy, schizophrenia, and long QT syndrome[58]. We compiled a subset of 111 and 489 unambiguously annotated benign and pathogenic KCN missense variants from ClinVar. On this variant set, COSMIS as a single metric showed strong performance (AUROC 0.854) (Fig. 8c). These results illustrate how COSMIS can be further applied to provide constraint maps in custom use cases and oligomeric structures beyond the precomputed scores we provide for 80.3% of proteins in the human proteome.

## Discussion

Establishing the clinical relevance of VUS is one of the biggest challenges to genomics-enabled precision medicine[59–63]. In this work, we hypothesized that patterns of genetic variation at neighboring sites in 3D space collectively reflect levels of functional constraint and that quantifying this constraint could aid VUS interpretation. We developed the COSMIS framework and

analyzed the distribution patterns of human genetic variants in the context of 3D protein structures. Our framework enabled us to map 3D mutational constraint at the resolution of individual sites in 80.3% of proteins in the human proteome. We further showed that our COSMIS score is accurate in predicting gene essentiality and variant pathogenicity and in aiding in the prioritization of de novo variants. Furthermore, it complements information provided by other commonly used metrics like phylogenetic conservation between species. The COSMIS framework is flexible and easily expanded to various applications as illustrated by our detection of constrained sites and pathogenicity predictions in ion channels using custom-built oligomeric homology models. We expect that our framework can be applied to a wider set of genes than analyzed in this work as the structural coverage of the human proteome and other species continue to expand.

Compared to existing constraint quantification approaches, our framework has several features that are particularly valuable for variant interpretation. First, our site-specific COSMIS score quantifies the variation in constraint at a finer scale than methods that generate a single score for an entire gene or subregions of a

gene. The COSMIS score is thus more precise and specific for interpreting missense variants than many common approaches. Second, our framework quantifies the constraint of sites in their 3D structural context. Compared to metrics that are based on 1D sequences, 3D structural context enabled the COSMIS score to capture native 3D interactions between residues that are far apart in sequence yet important for maintaining structural stability and functional integrity of the index site. In addition, the contact set filters out residues that are close in sequence but less likely to make contribution to the constraint of the site because they are distant in 3D. In fact, our 3D structure-based COSMIS score performed significantly better than the 1D sequence-based MTR score in predicting variant pathogenicity, while also providing important structural insights into the potential functional roles of constrained sites (Fig. 4b, c, and Supplementary Fig. 15). Third, our approach accounts for variation in mutation rates in the neutral model. This led to significantly better performance than a recent 3D-aware metric (MTR3D). Fourth, COSMIS provides scores for more than 80% of the human proteome by incorporating high-quality structural models from AlphaFold2, significantly more than previous structure-based analyses. Fifth, COSMIS can easily be applied to structures for new proteins or complexes, as illustrated on KCN genes. (COSMIS's strong performance on the KCN variants, suggests that it may be particularly well suited to pathogenicity prediction in ion channels.) Finally, COSMIS is complementary to other metrics. Combining the COSMIS score with phylogenetic conservation metrics yielded significantly higher performance than either approach alone in predicting variant pathogenicity. This suggests that future ensemble variant pathogenicity predictors may benefit from integrating 3D mutational constraint as quantified by the COSMIS score.

Our approach nevertheless has several limitations. First, the missense burden analysis and statistical identification of constrained contact sets is highly dependent upon the number and quality of variants used as references for the standing variation dataset. Current gnomAD samples carry only an average 6.3% and 10.3% of all possible missense and synonymous variants per contact set, respectively (Supplementary Fig. 16). As larger and more diverse reference genetic variation cohorts continue to increase the number of observed variants in each gene, even more accurate estimates of constraint on contact sets will be possible. It may also be possible to decrease the contact set distance threshold to capture more specific 3D interactions. Second, the COSMIS score does not directly consider the physicochemical severity of amino acid substitutions. While these substitution patterns likely contributed to the landscape of observed genetic variation and nucleotide mutability, explicit consideration of the severity of amino acid substitution could further improve estimates of site constraint. For instance, sites tolerant to both conservative and non-conservative substitutions are likely to be under less constraint than sites that are only tolerant to conservative substitutions. Third, while it is known that sites harboring variants with lower minor allele frequencies (MAFs) are likely under stronger selection pressure[64], as with previous approaches[1,12,14,16,17,20], we counted the number of unique variants observed at each site and did not explicitly account for their MAFs. Incorporation of MAFs and demographic structure into the formulation of scores such as the pLI[12], MTR[20], and COSMIS remains a promising topic. Finally, our analysis of the COSMIS scores of protein complexes is far from comprehensive, due to the lack of a proteome-wide structural database for human protein-protein interactions. Given the rapid advances in structure prediction for protein complexes[65–67], applying COSMIS to protein complexes at proteome-scale will soon be feasible. Together, we anticipate that our framework can be further improved in the future by

including larger human variation datasets, consideration of additional amino acid properties, and accounting for MAF and protein-protein interactions.

Looking forward, we anticipate that the structural landscape of constrained sites provided by COSMIS will facilitate prioritization of sites for mechanistic or functional investigation, especially those that have not been previously associated with clinically relevant phenotypes. For example, we have demonstrated that high-confidence constrained sites have a > 10-fold enrichment for pathogenic variants, yet 9404 out of the 10,955 proteins (85.8%) harboring at least one high-confidence constrained site lack any unambiguously annotated pathogenic variants in ClinVar. Variants at the constrained sites of some of these proteins are likely to be embryonic lethal, but many likely have pathogenic potential. Using COSMIS to guide investigation of the effects of variation at these sites on protein function will contribute useful insights into human health and disease.

## Methods

**Estimating mutability using the 1000 Genomes Project variant set**. We estimated sequence-context-dependent trinucleotide synonymous and missense mutability following previous procedures[12,14]. Briefly, we retrieved all single nucleotide variants from the 1000 Genomes Project Phase 3 variant set (2504 individual genomes)[11]. We filtered this initial set of variants to include only single nucleotide variants and excluded multiallelic variants, indels, and any variants with a filter tag other than "PASS". We focused on variants in intergenic regions obtained by excluding regions of the genome spanned by genes as annotated in GENCODE release 34[68]. We did not consider variants in the coding genome since they are enriched for purifying selection. For the entire intergenic genome, we counted every instance of each of the 64 trinucleotide sequences. We then identified all variable sites in the 1000 Genomes data with an annotated ancestral allele and assumed that each variable site represents a single ancestral mutation. To compute the probability of a trinucleotide $XYZ$ mutating to $XY'Z$, we divided the number of $Y \rightarrow Y'$ mutations in the context of $XYZ$ by the total number of occurrences of $XYZ$. As described in previous work[12,14], we scaled the probability by a proportionality constant[32] to derive the probability for one generation. In the end, we obtained a 64 by 3 matrix in which each row contains the probability of each of the three possible mutations of the central nucleotide of a given trinucleotide context. Protein-level mutability estimates obtained using our matrix agreed well with previous estimates (Supplementary Fig. 17). Our mutability table is available in the source code at our GitHub repository: https://github.com/CapraLab/cosmis. Our framework also enables the use of a custom mutation matrix.

**Mapping human reference proteome to Ensembl transcripts**. We started with the human reference proteome (UP000005640, UniProt release 2021_03), containing the reference amino acid sequences for a total of 20,600 proteins[69]. To determine whether the COSMIS scores for a protein can be computed, we first obtained the Ensembl stable transcript IDs for the protein through programmatic access of the UniProt database identifier mapping service (https://www.uniprot.org/help/api_idmapping). We used the transcript IDs as keys to extract coding sequence (CDS) from Ensembl CDS database. A valid CDS is necessary for the computation of COSMIS score because it is the basis for our mutation-probability-aware variant simulation procedure. A CDS is valid only if it begins with ATG, ends with a stop codon, and its translated amino acid sequence matches the UniProt reference sequence. We then used the transcript ID corresponding to the valid CDS as key to extract variant statistics from GRCh38-lifted gnomAD v.2.1.1. We used vcftools to remove all sites with a FILTER flag other than PASS from gnomAD v.2.1.1 and only kept single nucleotide variants. Completing this procedure for each reference protein resulted in a total of 16,533 proteins for which a "high-quality" protein 3D structure is also available in PDB, SWISS-MODEL repository, or AF2 database (see below).

**Per amino acid site synonymous and missense mutability of reference protein sequence**. We estimated the synonymous and missense mutability of each amino acid in the matched transcript of the protein in a nucleotide sequence context-dependent manner. In brief, the local trinucleotide sequence context was used to determine the mutability of each base in the coding region mutating to each other possible base and to determine the coding impact of each possible mutation. These mutability values were summed across the codon to determine its synonymous missense mutability. Specifically, for a given base in the codon, the trinucleotide sequence context is determined according to the coding sequence of the transcript as provided by the Ensembl CDS resource. The probability of the middle base mutating to one of the three other bases is queried in the mutation probability table and the type of change it would create is determined. The mutability is added to a running total for the type of mutation it would cause. This is repeated for the two

other possible mutations for every base in the codon. In the end, we obtained a pair of per-codon synonymous and missense mutability for each codon in each transcript.

**Estimating the per-transcript expected number of synonymous and missense variants.** We estimated the per-protein expected number of synonymous and missense variants through a fitted linear regression equation of per-protein total variant count on mutability. We first extracted 50,456 Ensembl transcripts for which SNVs were reported in gnomAD and whose CDS also met our criteria. For each of these transcripts, we then computed the synonymous and missense mutability of all codons and summed them to produce per-transcript total synonymous and missense mutability (Supplementary Data 11). Our transcript-level mutability estimates correlated strongly with previous estimates (Pearson's R of 0.94 and 0.95 for synonymous and missense, respectively; Supplementary Fig. 17)[14]. In parallel, we also tallied the total synonymous and missense variant counts reported in gnomAD for each of these transcripts. The total numbers of unique synonymous and missense variants from the 16,533 proteins studied in this work are 2.0 million and 4.1 million, respectively (these proteins are a subset of Supplementary Data 11). To establish the relationship between mutability and expected variant count under the null hypothesis of minimal constraint, we regressed the number of synonymous variants on the total synonymous mutability per transcript. As expected, and consistent with previous observations, we found that the total synonymous variant count can be accurately predicted by total synonymous mutability with a simple linear regression model ($\hat{y} = 6.42 \times 10^6 \times \mu - 0.18$, where $\mu$ is per-transcript total synonymous mutability, $R = 0.95$, Supplementary Fig. 18). As most synonymous variation is under minimal selective pressure, this model represents the relationship between mutability and observed variation when selection pressure is minimal. We thus estimated the expected count of missense variants $t_e$ under minimal selection for each of the transcripts by plugging the respective missense mutability into this regression model.

**Mapping transcripts to protein structures.** Computing the COSMIS score requires projecting missense variants onto 3D protein structures. We used the PDB as our primary source of protein structures. We used a summary table processed by SIFTS (https://www.ebi.ac.uk/pdbe/docs/sifts/quick.html, pdb_chain_uniprot.tsv.gz)[70] to obtain an one-on-one mapping between PDB chains and UniProt accession numbers. The PDB contains many cases where multiple PDB chains map to a single UniProt accession number. In these cases, we selected the PDB structure that has the most amino acid residues resolved. We also required PDB structures to have a resolution better than 5 Å and to cover as least one third of the reference amino acid sequence. If multiple PDB structures cover a protein sequence equally well, we selected the one that has the best resolution. We excluded structures for which sidechain coordinates are not resolved. For proteins for which no experimental structure in the PDB meets our criteria, we searched and retrieved structural models from the SWISS-MODEL repository (July 2021 release)[71] and the AlphaFold protein structure database (AF2)[38]. The SWISS-MODEL Repository is a database of annotated 3D protein structure models generated by the SWISS-MODEL homology-modeling pipeline[37]. The AF2 database is a collection of 3D protein structures for the human reference proteome predicted using the highly accurate AlphaFold2 method[39]. We first searched the SWISS-MODEL repository for models that have a sequence identity of at least 25% and cover at least one third of the amino acid sequence of the target sequence, in consideration of increasing the number of covered proteins and maintaining a reliable level of homology model quality[72]. In cases where multiple models satisfied these criteria, we selected the model with the highest sequence coverage to maximize the set of structure-mappable variants. For proteins with no homology models that meet our criteria, we relied on the AF2 structure database. In these cases, COSMIS scores were computed only if the AF2 predicted structure has at least one third of all residues predicted with a pLDDT > 50[39], and only such predicted residues were included in the computation of contact sets.

**Residue-level mapping between Ensembl transcript and protein structure.** The sequence of the experimental construct of a protein often does not match that of the reference sequence given in UniProt; e.g., the amino acid at position $i$ in the PDB file might be shifted by a few positions relative to its position in the translated peptide sequence of the corresponding transcript. We thus employed the SIFTS residue-level mapping resource[70] to maintain consistency between the UniProt and PDB residue numbering for each PDB chain-UniProt sequence pair. Specifically, for each PDB entry that was used as the 3D structure for a reference amino acid sequence, we downloaded the residue-level cross-reference data in XML format. Each of these XML file serve as the reference to ensure the accuracy of the mapping of each individual variant observed in gnomAD onto its location in the protein structure. Variants at positions that were not covered by protein structures were dropped.

**Construction of the COSMIS score.** For a protein sequence, the COSMIS score quantifies the constraint on a spatial region centered on each site of the sequence based on a reference protein structure. Construction of COSMIS is based on the concept of a contact set $S$. The contact set of site $r$, $S_r$, includes all sites in the

reference structure whose $C_\beta$ atoms (or $C_\alpha$ atoms in the case of glycines) are within 8 Å of the $C_\beta$ atom of $r$ and site $r$ itself. Conceptually, $S_r$ encloses the 3D spatial neighborhood surrounding site $r$ in the reference structure and typically also includes 2–3 residues that are sequence neighbors of site $r$ and important for determining the local secondary structure of $r$. But more importantly, $S_r$ also captures sites that are far apart in sequence but close in 3D space and most likely to contribute to the structural and functional integrity of $r$. For a protein that has $L$ amino acid sites, we thus have $L$ contact sets, one for each site. The 3D mutational constraint is quantified for each contact set and assigned to the site represented by the contact set.

We derived the COSMIS score numerically as follows. First, the site-level mapping between reference amino acid sequences and protein structures enables us to count the observed number of missense variants within each contact set. We designate this count $m_o$. Second, as the COSMIS score quantifies the deviation of $m_o$ from a null distribution of the number of missense variants within the contact set, we implemented a permutation-based simulation procedure to derive the null distribution. The simulation procedure starts with the computation of a normalized missense mutability of each codon in the transcript, that is

$$p_j = \frac{u_j}{\sum\limits_{i=1}^{L} u_i} \tag{1}$$

where $p_j$ is the missense mutability of codon $j$ normalized to the total missense mutability of the transcript, $u_j$ is the sequence context-dependent, unnormalized missense mutability of codon $j$ described in a previous section, and $L$ is the total number of amino acid sites in the protein sequence.

In each permutation, we then drew the missense variants from a multinomial distribution, where the number of trials is the total expected number of missense variants, $t_e$, for the protein and the probability for each amino acid site is $p_j$. We repeated this for $N = 10,000$ times. After each permutation, we count the number of missense variants sampled for each contact set. We denote the mean and standard deviation of this null distribution as $m_e$ and $m_\sigma$, respectively. The COSMIS score assigned to site $j$ is then computed as

$$COSMIS = \frac{m_o - m_e}{m_\sigma} \tag{2}$$

We also count the number of times out of the $N$ permutations the permuted count of missense variants in the contact set is less than or equal to the observed number $m_o$. We designate this count $K$. We calculated the empirical permutation $p$ value using the following formula:

$$p = \frac{(K + 1)}{(N + 1)} \tag{3}$$

**Gene lists.** We obtained the lists of genes of different levels of essentiality from https://github.com/macarthur-lab/gene_lists. These include: 294 haploinsufficient genes with sufficient evidence for dosage pathogenicity (level 3) as determined by the ClinGen Dosage Sensitivity Map as of Sep 13, 2018[43], 683 essential genes deemed essential in multiple cultured cell lines based on CRISPR/Cas screen data[45], 709 autosomal dominant disease genes from OMIM[46,47], 1183 autosomal recessive disease genes from OMIM[46,47], 913 nonessential genes deemed nonessential in multiple cultured cell lines based on CRISPR/Cas screen data[45], and 284 olfactory receptors[44]. Several of these lists were also previously used to benchmark the pLI metric that quantifies intolerance to functional variation[12]. Genes in these lists are identified by their HGNC symbols. To link with COSMIS scores (indexed by UniProt access numbers), we mapped HGNC symbols to UniProt accession numbers through programmatic access of the UniProt database identifier mapping service (https://www.uniprot.org/help/api_idmapping). Collectively, our framework provides scores for 213 haploinsufficient, 622 essential, 584 dominant, 999 recessive, 721 nonessential, and 284 olfactory genes. These gene lists are available at our GitHub repository and also as Source Data files for Fig. 5c.

**Intra- and inter-species constraint metrics.** We focused our comparison of COSMIS with other evolutionary constraint metrics rather than ensemble variant effect prediction methods derived through machine-learning or score aggregation. In particular, our primary interest was to compare COSMIS with recently developed human-variation-based constraint metrics, i.e., the residual variation intolerance score (RVIS)[13], the missense Z score[12,14], the probability of loss-of-function intolerance metric (pLI)[12], missense tolerance ratio (MTR)[20], and missense tolerance ratio 3D (MTR3D)[26]. Additionally, we compared COSMIS with GERP++[73], phyloP[74], phastCons[75], and ConSurf[76] to investigate the potential synergistic effects of combining intra- and inter-species metrics for predicting variant pathogenicity. We computed the ConSurf scores using the Rate4Site program[3] with default parameters and no branch length optimization. We obtained the 100-way multiple sequence alignment for each of the proteins and the tree file (hg38.100way.nh [http://hgdownload.cse.ucsc.edu/goldenpath/hg38/multiz100way/hg38.100way.nh]) from the UCSC Genome Browser. We computed relative solvent accessibility using DSSP 3.0[77] within the Biopython framework[78]. The sources of other scores were listed in Supplementary Data 12. The subset of

amino acid sites in the human reference proteome for which all scores are available can be found at our GitHub repository and also as Source Data files for Fig. 6b.

**COSMIS score distribution of ClinVar variants**. We evaluated the ability of COSMIS to predict pathogenicity of missense variants using the ClinVar variant resource[79] (retrieved in August 2021) as an evaluation set. Our evaluation set consisted of solely ClinVar missense variants that were labeled as "Pathogenic", "Likely pathogenic", or "Pathogenic/Likely pathogenic" for true positive (pathogenic) variants and "Benign", "Likely benign", or "Benign/Likely benign" for true negative (benign) variants. The VUS set consisted of variants labeled as "Uncertain significance". All variants were required to have a review status of at least one star and no conflicting interpretation. Due to the dependency of COSMIS score on 3D structures, we also required variants in the evaluation set to be mappable to our 3D structure sets. Any ClinVar variant designated as "no assertion criteria provided", "no assertion provided", "no interpretation for the single variant", or not covered by protein structures was excluded from the evaluation set. Collectively, these restrictions resulted in 19,346 benign, 14,824 pathogenic, and 115,172 VUS sites for which the COSMIS score can be computed. The actual number of variants is higher than the respective number of unique sites because some sites can be hotspots where multiple variants have been reported. These variants and their COSMIS scores are available in Supplementary Data 3. By removing the proteins with pathogenic variants in this dataset from the list of 10,955 proteins with at least one high-confidence constrained site, we obtained 9404 proteins for which no unambiguous pathogenic variants have been reported in ClinVar. For variant labeling, we also note that in the analysis of score distributions for sub-groups (Supplementary Fig. 9), variants labeled "Pathogenic/Likely pathogenic" or "Benign/Likely benign" were excluded. ORs for each percentile bin were calculated by $OR = \frac{a/b}{c/d}$, where $a$ is the number of pathogenic variants in a bin, $b$ is the number of pathogenic variants not in the bin, $c$ is the number of benign variants in the bin, and $d$ is the number of benign variants not in the bin. We calculated 95% percent CIs from the standard error, $s.e. = \sqrt{\frac{1}{a} + \frac{1}{b} + \frac{1}{c} + \frac{1}{d}}$. The lower bound of the CI is calculated using the expression $e^{\ln[OR] - 1.96 \times s.e.}$, and the upper bound of the CI is calculated by $e^{\ln[OR] + 1.96 \times s.e.}$.

**COSMIS score distribution of de novo missense variants**. The set of de novo missense variants was obtained from a previous analysis[51]. This set consists of 5113 de novo missense variants in 5620 neurodevelopmental disorder probands ("case" variants) and 1269 de novo missense variants in 2078 unaffected siblings of autism spectrum disorder probands ("control" variants). Following the procedure of residue-level mapping, we were able to map 2271 case variants and 541 control variants to protein 3D structures and compute the COSMIS scores for these variants. These variants and their COSMIS scores are available in Supplementary Data 8. ORs and CIs for each percentile bin were calculated in a similar manner as with ClinVar variants, where $a$ is the number of case variants in a bin, $b$ is the number of case variants not in the bin, $c$ is the number of control variants in the bin, and $d$ is the number of control variants not in the bin.

**Application of COSMIS to potassium ion channels**. We selected a set of 41 clinically relevant potassium ion channels for which missense variants have been unambiguously annotated in ClinVar following the same procedure as stated in the previous section on curating ClinVar variants. Experimental structures were available in the PDB (as of Dec. 2020) for ion channels encoded by the *KCNH2*, *KCNJ11*, *KCNQ1*, *KCNQ2*, and *KCNQ4* genes. For the remaining 36 potassium channels, we leveraged their high sequence identity (mean sequence identities between template and target ion channels are 56.6%) to the potassium channels with available structures and constructed homology models in their oligomeric states using the SWISS-MODEL interactive workspace[37]. We removed residues with a QMEAN score of <0.3 in the intracellular intrinsically disordered regions of these ion channels. More information about templates used in our homology modeling and evaluations of model qualities can be found in Supplementary Data 9. Using these potassium channel 3D structures, we computed two sets of COSMIS scores, based on monomers and oligomers, respectively, for 4762 interface and 14,331 non-interface sites. A site is considered part of the interface if the number of sites in its contact set is larger in the oligomer than in the monomer. Collectively, we were able to map and compute the COSMIS scores for a total of 111 and 489 unambiguously annotated benign and pathogenic KCN missense variants from ClinVar. Our ion channel variant dataset, together with their precomputed COSMIS scores, are available at our GitHub repository and also as Source Data files for Fig. 8.

**Application of COSMIS to homodimeric structures from the PDB**. We obtained a set of homodimeric proteins from the Interactome INSIDER resource[80]. Briefly, we first retrieved all "highest-confidence" protein-protein interfaces in human from http://interactomeinsider.yulab.org (accessed in Jan. 2022). We then excluded interfaces that were not calculated from PDB structures or consist of two different protein subunits. For the remaining homodimeric interfaces, we mapped their UniProt accession numbers to PDB chains using a summary table processed by SIFTS (https://www.ebi.ac.uk/pdbe/docs/sifts/quick.html, pdb_chain_uniprot.tsv.gz)[70].

We retained proteins that were mapped to PDB files consisting of two and only two identical subunits. This step excluded homodimeric interfaces that are part of higher-order protein complexes. In summary, we computed the COSMIS scores for 1678 proteins (Supplementary Data 13) using both monomeric and homodimeric structures from the PDB.

**Reporting summary**. Further information on research design is available in the Nature Research Reporting Summary linked to this article.

## Data availability

Source data are provided with this paper, i.e., the code and source data for reproducing all graphs in the main text figures and supplementary figures (except Fig. 1, which is a schematic with no source data) are available in the Source Data file [https://figshare.com/articles/dataset/Source_Data/19742404]. Precomputed COSMIS scores for 16,533 proteins from the UniProt human reference proteome can be downloaded at https://github.com/CapraLab/cosmis. We have also created a web application for interactive exploration of COSMIS scores in the context of protein 3D structures. The web application is available at https://cosmis-app.herokuapp.com/.

The following publicly available datasets and databases were used:
1000 Genomes phase 3: http://ftp.1000genomes.ebi.ac.uk/vol1/ftp/release/20130502/
AlphaFold2: https://ftp.ebi.ac.uk/pub/databases/alphafold/latest/UP000005640_9606_HUMAN_v2.tar
ClinVar (retrieved in August 2021): https://ftp.ncbi.nlm.nih.gov/pub/clinvar/vcf_GRCh38/
GECODE (release 34): http://ftp.ebi.ac.uk/pub/databases/gencode/Gencode_human/
Gene lists: https://github.com/macarthur-lab/gene_lists
gnomAD v2.1.1: https://gnomad.broadinstitute.org/
INSIDER (accessed in Jan. 2022): http://interactomeinsider.yulab.org
PDB: https://www.rcsb.org/
SIFTS: https://www.ebi.ac.uk/pdbe/docs/sifts/quick.html
SWISS-MODEL: https://swissmodel.expasy.org/repository/species/9606
UniProt: https://www.uniprot.org/help/uniprotkb
UniProt human reference proteome (release 2021_03): https://www.uniprot.org/proteomes/UP000005640

## Code availability

In addition to the precomputed scores, COSMIS can also be downloaded and run as a standalone application locally. The source code of COSMIS is publicly and freely available at https://github.com/CapraLab/cosmis. The source code for reproducing all graphs in the main text figures and supplementary figures (except Fig. 1, which is a schematic with no associated source code) is available at https://figshare.com/articles/dataset/Source_Data/19742404.

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

## Acknowledgements

This research was funded by the National Institute of Health grants R35GM127087 (J.A.C.) and R01LM013434 (J.A.C.) and American Heart Association fellowship 20POST35220002 (B.L.). This work was conducted in part using the resources of the Advanced Computing Center for Research and Education at Vanderbilt University, Nashville, TN. We thank Dr. David Rinker for constructive criticism of the manuscript, Dr. Evonne McArthur for help with the layout of the web application, and members of the Capra Lab for insightful discussions.

## Author contributions

B.L. and J.A.C. conceived the study. B.L. compiled the datasets, wrote the source code, and analyzed the data. B.L., D.M.R., and J.A.C. interpreted the results. B.L. and J.A.C. wrote the manuscript with edits from D.M.R. B.L. and J.A.C. acquired funding to support this work. D.M.R. and J.A.C. supervised the project. All authors approved the submitted version of the manuscript.

## Competing interests

The authors declare no competing interests.
