## [Peer Review File · Nature Communications]

REVIEWER COMMENTS

Reviewer #1 (Remarks to the Author):

Review of Li et al “The 3D spatial constraint on 6.1 million amino acid sites in the human proteome”

This paper by Li et al. strives for a systematic assessment of evolutionarily constrained sites in the human proteome. The novelty is in combining existing approaches of variant mapping to the computational protein structures available through AlphaFold2, SWISS-MODEL, and PDB - using a statistical model to quantify depletion of population variants (gnomAD) at index sites. The authors well reference similar 3D variant mapping approaches, which however typically are used for cancer variant prioritisation. Pathological variants are then put into context with the constrained sites discovered. The main problem seems to be that the generation of the null model could be impacted by insufficient variant annotation, which is something recognized and discuss by authors themselves.

The authors hypothesize that 3D neighbouring amino acid sites (“contact sets”) collectively shape the level of constraint on each site and discuss implications for pathogenicity and functional understanding of variants. They drive a COSMIS score which essentially is a Z-score of variant frequency against an empirical random distribution; generated from drawing of missense variants from a multinomial distribution. As the authors note, this depends strongly on the reference data. Comparisons to other variant predication tools is reported in Figure 6/7.

Key findings include 10,955 proteins with an average of 28.6 high-confidence constrained sites per protein, which are 13.5-fold enriched for pathogenic variants, however most proteins lack annotated pathogenic variants in ClinVar. Particular cases: 72 proteins with more than 50% high-confidence constrained sites under extreme purifying selection. With the oligomeric potassium channel 3D structures examples, their results also underline the importance of protein interaction interfaces. also identified in recent studies on variation effects on protein-protein interactions.

In general, the comments on COSMIS’s limitations are excellent.

Points for consideration:

- *) How much impact does the uncertainties in the variants in ClinVar have on the calculation of the COSMIS score, is it robust to perturbation?
- *) Though this is discussed to some extend: What happens if the distance constrains are smaller than 8 Angstrom?
- *) Impact of constrained sites located in protein interaction interfaces is, except for the potassium channel example, not address. However, because being so prominent in the PPI examples, a discussion to extending this approach towards PPI interfaces (essentially residues in spatial proximity, just form a second peptide chain) could be interesting.

Reviewer #2 (Remarks to the Author):

The authors of the manuscript “The 3D spatial constraint on 6.1 million amino acid sites in the human proteome” by Li, Roden, & Capra describe a novel method to prioritize variants in human disease studies. The main innovations are (i) consideration of a larger number of protein structures than prior art (thanks to AlphaFold) and (ii) modeling the dependence of mutation rate on local sequence context. Approaches such as these that leverage orthogonal and ever-larger sources of information are needed in the field.

The motivation for this work is sound and the methodology here is innovative, as it uses an unprecedented number of reference proteins; calculates protein-level site specific and 3D constraint metrics; incorporates 3D constraint with interspecies and intraspecies for more accurate predictions of constraint. Several previous research efforts focused on identifying genes and subgene regions subject to purifying selection, or constraint, in the general human population. To date, a majority of these efforts have examined constraint acting on protein-coding sequences in 1D space - that is, the raw coding sequence. However, given the complex structural biology underlying protein folding and function, examination of constraint in 3D space at the protein level can identify 1D distant loci essential for proper protein function. Therefore, Capra et al leveraged an unprecedented amount of protein structure information from the PDB, SWISS-MODEL repository, and AlphaFold2 database to examine constraint at the level of individual amino acid sites and 3D amino acid interactions across monomeric and oligomeric proteins. To this end, the authors developed a COSMIS metric that accounts for the mutability of different trinucleotides. This framework led to the discovery that 50% of amino acids 15 residues away from a central AA are not in close 3D contact. Further, their COSMIS methodology allowed for improved accuracy to detect ClinVar-defined pathogenic variants when coupled with inter- and intra-species constraint information. Therefore, this work has significant implications for informing future research efforts on the functional importance and potential pathogenicity of VUS that coincide with highly constrained sites and 3D regions.

There are some concerns or areas of confusion that I suggest be addressed.

There is a lack of clarity about the datasets used to generate observed and expected mutation counts in the results section. The main text says gnomAD but methods say 1000 Genomes Project.

Is the comparison of intraspecies and interspecies constraint scores necessary? It didn't seem to provide anything novel — similar methods produce similar results — it is unclear how this alone led to their justification for “integrating interspecies scores with COSMIS”

“We hypothesized ... quantifying [3D] constraint could aid VUS interpretation”. It is not clear that the authors showed an example of this. If it is their hypothesis, don't they have to have an example where a VUS was successfully “rescued” and identified as a pathogenic variant?

One section in the Results is entitled "COSMIS improves the interpretation of de novo variants in neurodevelopmental disorders". I don't see how. The authors show that COSMIS scores for de novo mutations in cases are on average significantly different from those for de novo mutations in healthy siblings. This suggests to me that a significant number of the de novo mutations in the affected sibling are contributing to their disease, but doesn't tell us which ones. Of course, one could rank the de novo mutations in the affected sibling by COSMIS score, but there is no truth set to gauge whether that is any better than choosing a random ranking of the de novo mutations in the affected sibling. In short, I think this section argues that de novo mutations collectively contribute to this disorder, but doesn't convince us that using any metric (COSMIS or otherwise) can reliably prioritize those mutations for further investigation. I would suggest renaming the section accordingly.

COSMIS predicts greater constraint when the 3D template of a test protein (KCN) is chosen to be its oligomeric form instead of its monomeric form.

The authors use the term "custom-built" a lot, without defining what is actually “built”. I would request a sentence explaining what was built and why.

Perhaps the authors could generalize this result by using RoseTTA Fold to compute predicted protein-protein complexes for other proteins, or by using whatever protein-protein complexes are reported in the RoseTTA Fold paper. <https://www.science.org/doi/10.1126/science.abj8754>

Minor Concerns:

No discussion of gnomAD versions used

The authors show they predict protein constraint in oligomeric states but don't use RoseTTAfold which is supposed to be better at this. Is this an opportunity for future improvement

“With an average of 28.6 high-confidence constrained sites per protein”  are these constrained sites clustered together in 3D space?

Discussion section: “while also providing important biophysical insights into the potential functional roles of constrained sites”. Please provide a concrete example of this

Response to Referees

We thank the editor and reviewers for evaluating our manuscript, “The 3D spatial constraint on 6.1 million amino acid sites in the human proteome” (NCOMMS-21-44162) for publication in *Nature Communications*. We were pleased that the reviewers appreciated the contribution of our work and are grateful for their suggestions. We respond to each of their comments in this document. We believe that the new manuscript is substantially improved and addresses the reviewers’ concerns.

Reviewer #1:

**) How much impact does the uncertainties in the variants in ClinVar have on the calculation of the COSMIS score, is it robust to perturbation?*

Response: We thank the reviewer for this important question. We would like to clarify that our source of variants for computing the COSMIS score is gnomAD v 2.1.1. Variants in ClinVar were not used for computing the COSMIS score; however, we anticipate that on average COSMIS scores should reflect the uncertainties in pathogenicity annotations among ClinVar variants. To confirm this, we computed the COSMIS score distribution for variants of uncertain significance (VUS) from ClinVar. The median COSMIS score for VUS is -0.31 (compared to 0.0 and -1.1 for benign and pathogenic variants, respectively). This is consistent with the expectation that VUS contain a mixture of variants with various functional effects, but are overall less constrained than those labeled as pathogenic. We further analyzed the score distributions of benign, likely benign, likely pathogenic, and pathogenic variants separately. We have added the results of these analyses to the revised manuscript. We copy the relevant sections and figures below, with new text and results highlighted in red.

Revision:

To quantify the ability of COSMIS to contribute to identification of disease-associated protein variants, we compared the COSMIS scores for a total of 19,346 benign and 14,824 pathogenic missense variants with unambiguous annotations of clinical significance and 115,172 VUS from ClinVar (Methods, Supplementary Table 3). Pathogenic variants have a significantly lower COSMIS score distribution than benign variants (median -1.1 vs. -0.0, respectively; $p < 2.2 \times 10^{-308}$, two-sided Mann-Whitney U test; Fig. 5a). **The median COSMIS score of the VUS set is -0.31, consistent with the expectation that it is a mixture of variants of various functional effects. The distance threshold for defining residue contact has little effect on the score distributions of these variant sets relative to each other (Supplementary Fig. 8). Further division of variants into four subgroups, i.e., benign, likely benign, likely pathogenic, and pathogenic, shows that the median score of likely benign variants is slightly lower than that of benign variants (-0.12 vs. 0.07; $p = 3.3 \times 10^{-23}$, two-sided Mann-Whitney U test), whereas pathogenic and likely pathogenic variants both have lower scores (median -1.12 vs. -1.17, respectively; $p = 0.01$, two-sided Mann-Whitney U test) (Supplementary Fig. 9). Collectively, the significant negative shift for pathogenic variants suggests strong constraint in their spatial neighborhoods, while the average**

neutral COSMIS score of benign variants suggests less constraint on missense variants in their contact sets.

Fig 5. The COSMIS score is strongly correlated with both pathogenicity and gene constraint level. **a)** COSMIS score distributions for 19,346 benign, 14,824 pathogenic, and 115,172 VUS from ClinVar (Methods). Pathogenic variants have significantly more constrained 3D spatial neighborhoods (COSMIS score median -1.1) than benign variants (median score 0.0) ($p < 2.2 \times 10^{-308}$, two-sided Mann Whitney U test). **b)** Odds ratio (OR) of ClinVar pathogenic variants versus benign variants for different COSMIS score percentile bins (lower bins correspond to more constrained COSMIS scores). Amino acid sites with lower COSMIS scores are enriched for pathogenic variants whereas sites with higher scores are depleted of pathogenic variants. The horizontal dashed line indicates OR = 1. **c)** COSMIS score distributions of amino acid sites in six groups of genes with different functional annotations (and the dataset as a whole). As the anticipated functional constraint on each category increases (top-to-bottom), amino acid sites in genes in the category have more constrained COSMIS scores on average.

Fig S9. Distributions of COSMIS scores for subgroups of variants from ClinVar. Further division of variants into four subgroups, i.e., benign, likely benign, likely pathogenic, and pathogenic, shows that the median score of likely benign variants is slightly lower than that of benign variants (-0.12 vs. 0.07; $p = 3.3 \times 10^{-23}$, two-sided Mann-Whitney U test), whereas little difference exists between pathogenic and likely pathogenic variants (median -1.12 vs. -1.17, respectively; $p = 0.01$, two-sided Mann-Whitney U test).

*) *Though this is discussed to some extent: What happens if the distance constraints are smaller than 8 Angstroms?*

Response: We thank the reviewer for asking for this important clarification. We chose 8 Å between the C_{β} atoms of two residues as the distance threshold because this is a well-accepted empirical parameter for defining two residues in contact (Shrestha et al., 2019). To further explore the effects of distance threshold, we have run two additional parallel analyses in which the distance threshold was set to 6 Å and 10 Å, respectively. Our analyses indicate that the distance threshold has little effect on the behavior of COSMIS score or our main conclusions. We copy the relevant sections and figures below, with new text and results highlighted in red.

Revision:

Fig S4. Statistics of long-range contacts (separated by more than 15 residues along the 1D sequence) at 6 Å and 10 Å distance thresholds. **a, b)** Contact sets capture long-range sites that interact in 3D at 6 Å and 10 Å distance thresholds, respectively. The bar plots show the fraction of all 6.1 million sites with at least a certain fraction of long-range 3D contacts in their contact sets. **c, d)** Similar to Fig. 2d, but at 6 Å and 10 Å distance thresholds, respectively. The bar plots show the fraction of all 6.1 million sites that have at least a certain fraction of 1D sequence neighbors that do not form 3D contacts.

Fig S5. Distribution of the COSMIS scores for 6.1 million unique amino acid sites of the reference human proteome, computed at 6 Å (a) and 10 Å (b) distance thresholds, respectively. As with the score distribution computed at 8 Å distance threshold, an average amino acid site in the human proteome is depleted of missense variants in its contact set (median COSMIS scores are -0.498 and -0.554 at 6 Å and 10 Å distance thresholds, respectively).

Fig S8. Distributions of COSMIS scores computed at 6 Å (a) and 10 Å (b) for ClinVar variant sets. a) Median scores of benign, VUS, and pathogenic variants using a 6 Å threshold to define contact sets are 0.10, -0.22, and -0.87, respectively. a) Median scores of benign, VUS, and pathogenic variants using a 10 Å threshold to define contact sets are -0.10, -0.39, and -1.29, respectively. All p values were computed from two-sided Mann Whitney U tests.

*) *Impact of constrained sites located in protein interaction interfaces is, except for the potassium channel example, not addressed. However, because being so prominent in the PPI examples, a discussion to extending this approach towards PPI interfaces (essentially residues in spatial proximity, just form a second peptide chain) could be interesting.*

Response: We agree on the importance of protein-protein interfaces and the possibility of extending COSIMS to a broader set of protein complexes. To this end, we have extended our analysis of oligomeric structures by incorporating a larger set of 1,678 human homodimeric proteins from the PDB. We copy the relevant sections and figures below, with new text and results highlighted in red.

Revision:

Application of COSMIS to homodimeric structures from the PDB

We obtained a set of homodimeric proteins from the Interactome INSIDER resource (Meyer et al., 2018). Briefly, we first retrieved all “highest-confidence” protein-protein interfaces in human from <http://interactomeinsider.yulab.org> (accessed in Jan. 2022). We then excluded interfaces that were not calculated from PDB structures or consist of two different protein subunits. For the remaining homodimeric interfaces, we mapped their UniProt accession numbers to PDB chains using a summary table processed by SIFTS (<https://www.ebi.ac.uk/pdbe/docs/sifts/quick.html>, pdbe_chain_uniprot.tsv.gz) (Dana et al., 2019). We retained proteins that were mapped to PDB files consisting of two and only two identical subunits. This step excluded homodimeric interfaces that are part of higher-order protein complexes. In summary, we computed the COSMIS scores for 1,678 proteins (Supplementary Table 13) using both monomeric and homodimeric structures from the PDB.

Fig S14. Applying COSMIS to homodimeric structures. a) COSMIS score distributions for interface and non-interface amino acid sites in 1,678 homodimeric protein structures from the PDB. Overall, interface sites involved in oligomerization (making more 3D contacts in oligomers than in monomers) have significantly lower COSMIS scores than non-interface sites (median -0.64 vs. -0.55, $p = 7.4 \times 10^{-54}$, two-sided Mann-Whitney U test). b) COSMIS scores of interface sites computed based on homodimers are generally lower than those computed based on monomers (median difference -0.11, $p = 3.1 \times 10^{-12}$, two-sided Mann-Whitney U test).

Reviewer #2:

1. *There is a lack of clarity about the datasets used to generate observed and expected mutation counts in the results section. The main text says gnomAD but methods say 1000 Genomes Project.*

Response: We thank you for pointing out the need to clarify our description of what datasets were used to generate observed and expected variant counts. In the revised manuscript, we have clarified that observed variant counts were derived from gnomAD v2.1.1., and that the expected variants counts were derived by a simulation procedure that accounts for sequence-context aware mutation probabilities. These mutation probabilities were estimated using the 1000 Genome Project variant data. We copied the relevant sections and figures below, with new text and results highlighted in red.

Revision:

The COSMIS framework maps spatial constraint on proteins in high resolution

We developed the COSMIS framework to quantify the 3D spatial constraint at each site in a protein structure **by analyzing the patterns of genetic variants from large-scale sequencing projects in the context of protein structures (Fig. 1 and Methods)**. Our framework estimates the constraint on a site of interest (index site) as the depletion of missense variants in its **3D spatial neighborhood (i.e., contact set, see definition below)** compared to the number expected if it were evolving neutrally. We quantify this as the deviation of observed count of missense variants (m_o) from the **expected count** (m_e , accounting for transcript and codon missense mutability), divided by the standard deviation of the expected distribution (m_σ). We designate this Z score as the COSMIS score and assign it to the index site. **We obtain the observed variant count in each contact set by mapping variants catalogued by the Genome Aggregation Database (gnomAD, v2.1.1) (Karczewski et al., 2020) onto protein structures. To obtain the expected variant count distribution, we use a procedure that simulates mutation under neutral evolution based on a 64 by 3 sequence-context-dependent mutability matrix derived from the 1000 Genomes Project variant set (Auton et al., 2015).** We also compute an empirical p-value for each COSMIS score based on the simulation procedure and the resulting expected count distribution (Methods). **According to our formulation of the score**, a lower value indicates a greater depletion of missense variants in the spatial neighborhood and hence lower missense variation tolerance.

2. *Is the comparison of intraspecies and interspecies constraint scores necessary? It didn't seem to provide anything novel — similar methods produce similar results — it is unclear how this alone led to their justification for “integrating interspecies scores with COSMIS”*

Response: We thank the reviewer for this critical question. We would like to clarify that interspecies metrics have relatively low correlation with metrics in the intraspecies group (Fig. 6b). However, both groups of metrics demonstrated various levels of predictive ability for variant pathogenicity (Fig. 6a and Supplementary Fig. 12). This suggests that these two groups of metrics contain some complementary information that, when combined, could improve the performance of predicting variant pathogenicity. This idea was confirmed in Fig. 6c and 6d, i.e., combining

COSMIS (best intraspecies metric) and ConSurf (best interspecies metric) outperformed either alone. We have revised the text to make this clarification. We copy the relevant section below, with new text highlighted in red.

Revision:

To illustrate the differences between the intraspecies constraint scores and interspecies phylogenetic conservation metrics, we additionally computed the correlations for four common interspecies phylogenetic conservation metrics (GERP++, phyloP, phastCons, ConSurf). Phylogenetic conservation metrics are generally more correlated with each other than with any of the intraspecies constraint scores (Fig. 6b). For example, the lowest Spearman's ρ between the four phylogenetic conservation metrics is 0.47 (GERP++ vs. phyloP), higher than the highest Spearman's ρ between a phylogenetic metric and an intraspecies constraint score (i.e., 0.38, ConSurf vs. RVIS and Missense_Z). This is consistent with previous finding that intraspecies constraint metrics are only modestly correlated with phylogenetic conservation (Gussow et al., 2016; Havrilla et al., 2019). **Together with the observation that both groups of metrics demonstrated predictive ability for variant pathogenicity (Fig. 6a and Supplementary Fig. 12), this suggests that these two groups of metrics contain complementary information that, when combined, could improve prediction of variant pathogenicity.**

3. *"We hypothesized ... quantifying [3D] constraint could aid VUS interpretation". It is not clear that the authors showed an example of this. If it is their hypothesis, don't they have to have an example where a VUS was successfully "rescued" and identified as a pathogenic variant?*

Response: In addition to our demonstration that COSMIS is significantly predictive of pathogenicity (Figs. 5 and 6), we agree that it is important to demonstrate the ability of COSMIS to "rescue" specific pathogenic variants, when comparing to approaches that do not leverage 3D structure information. In the revised manuscript, we provide an example analysis using COSMIS scores to interpret a set of functionally characterized VUS in the SCN5A sodium channel. Our analysis shows that COSMIS "rescues" pathogenic variants that would be misclassified by the corresponding sequence-only constraint metric, MTR. We copy the relevant sections and figures below, with new text and results highlighted in red.

Revision:

COSMIS complements existing quantifications of intra- and inter-species constraint

To assess the relationship between COSMIS and other intra- and interspecies constraint metrics, we first compared COSMIS to four commonly used intraspecies constraint metrics that do not consider structural context (MTR, RVIS, pLI, and Missense_Z). We compared these other metrics to COSMIS in their ability to identify pathogenic variants using a total of 8,062 benign and 7,256 pathogenic missense variants from ClinVar for which all scores could be computed (Supplementary Table 6). COSMIS achieved a significantly higher area under the receiver operating characteristic curve (AUROC) than the other intraspecies constraint metrics (e.g., 0.733 vs. 0.653 for COSMIS vs. MTR, $p = 1.0 \times 10^{-65}$, two-sided DeLong's test, Fig. 6a). **To illustrate these patterns, analysis of the COSMIS scores for a set of functionally characterized VUS in the SCN5A sodium channel (Glazer et al., 2020) shows that COSMIS "rescues" pathogenic variants**

that would be misclassified by MTR (Supplementary Fig. 11; Supplementary Table 7). These results suggest that 3D neighboring residues contribute critical information about the functional importance of index sites. We additionally compared COSMIS to a recently developed version of MTR that considers missense variants in 3D neighborhoods (MTR3D), but does not account for sequence context-dependent mutability (Perszyk et al., 2021; Silk et al., 2021). COSMIS also performs significantly better than MTR3D (i.e., 0.733 vs. 0.665, $p = 2.5 \times 10^{-50}$, two-sided DeLong's test, Fig. 6a), suggesting that accounting for the variability of mutability is essential to estimate constraint.

Fig S11. Applying COSMIS to classify the pathogenicity of SCN5A variants of uncertain significance (VUS). **a**) To demonstrate the utility of COSMIS in real-world interpretation of VUS, we computed the ROC curves of COSMIS and MTR based on the 44 SCN5A VUS reclassified in Glazer *et al.* using a high-throughput functional assay (12 benign/likely benign and 32 pathogenic/likely pathogenic; Supplementary Table 7). COSMIS performs substantially better than MTR, illustrating the benefits of incorporating 3D structural context. **b**) Our analysis further highlights five SCN5A VUS (W879R, W1345C, L1346P, E1574K, and N1722D) that are ranked among the top 10 variants most likely to be pathogenic by COSMIS. None of these variants is ranked among the top 10 by MTR (Supplementary Table 7). These VUS were classified as likely to be pathogenic according to functional data reported in Glazer *et al.* (peak current density shown as % of wild type in parentheses). The SCN5A structure is rendered in cartoon and colored by domain (PDB ID: 6LQA). Sites where these five variants are located are rendered in spheres.

4. One section in the Results is entitled "COSMIS improves the interpretation of de novo variants in neurodevelopmental disorders". I don't see how. The authors show that COSMIS scores for de novo mutations in cases are on average significantly different from those for de novo mutations in healthy siblings. This suggests to me that a significant number of the de novo mutations in the affected sibling are contributing to their disease, but doesn't tell us which ones. Of course, one could rank the de novo mutations in the affected sibling by COSMIS score, but there is no truth set to gauge whether that is any better than choosing a random ranking of the de novo mutations in the affected sibling. In short, I think this section argues that de novo mutations collectively contribute to this disorder, but doesn't convince us that using any metric (COSMIS or otherwise) can reliably prioritize those mutations for further investigation. I would suggest renaming the section accordingly.

Response: We agree with the reviewer on this thoughtful comment. In the revised manuscript, we changed the title to “COSMIS distinguishes *de novo* variants in neurodevelopmental disease cases and controls”. In addition, we also agree that our analysis indicates that current metrics (including COSMIS) still can’t reliably prioritize *de novo* mutations for further investigation. We copy the relevant sections and figures below, with new text and results highlighted in red.

Revision:

For context, we compared the ability of COSMIS to enrich for case variants with the other inter- and intraspecies metrics considered previously. We did this analysis using 1,506 case and 306 control variants for which all 10 scores are available (Supplementary Table 9). For COSMIS, 24.2% case and 8.2% control variants fall within the 10th percentile of most constrained sites (i.e., COSMIS score < -1.85), corresponding to an odds ratio (OR) of 3.6 ($p = 2.6 \times 10^{-11}$, two-sided Fisher’s exact test) (Fig. 7b). Both MTR and pLI achieved the next highest OR of 2.6 at the 10th percentile of most constrained sites, while being lower than COSMIS. Except for MTR3D, which has an OR of 2.1, the ORs of all the rest metrics are below 2 (Fig. 7b). **COSMIS also has the highest ORs at other thresholds (5th and 20th percentiles) (Supplementary Fig. 13). The modest performance of all evaluated metrics (including COSMIS) is not surprising as we do not expect all *de novo* variants in cases to be causal/pathogenic (Paludan-Muller et al., 2017). To more accurately benchmark the ability of these metrics to predict the pathogenicity of specific *de novo* variants, a well-established set of *de novo* variants with clinically validated disease associations is needed. Nevertheless, our analysis indicates that reliably prioritizing *de novo* variants for further investigation is still a challenging problem for contemporary metrics, as has been previously suggested (Havrilla et al., 2019; Pejaver et al., 2020).**

5. *COSMIS predicts greater constraint when the 3D template of a test protein (KCN) is chosen to be its oligomeric form instead of its monomeric form. The authors use the term "custom-built" a lot, without defining what is actually "built". I would request a sentence explaining what was built and why.*

Response: Thank you for pointing out the need for further clarification of the term “custom-built”. We use this term to refer to a protein structure that is not already available in public protein structure databases. Such “custom-built” 3D structures for single proteins or protein-protein complexes can be constructed using macromolecular modeling software packages. While we have precomputed COSMIS scores systematically for >80% human proteins in an automated fashion using available models, it is conceivable that a protein of interest might be in an alternative conformation or be part of a protein-protein interaction whose structures are not available in public structure databases. In this section, we are illustrating the flexibility of our framework to work with protein structures constructed outside of public structure databases and provide novel insights. In the revised manuscript, we have made the clarification.

Revision:

Applying COSMIS to custom-built oligomeric potassium channel structures

While we have precomputed COSMIS scores for >80% human proteins using publicly available structures, it is conceivable that the structure of a protein of interest might not be available in

public databases. To demonstrate the flexibility of our framework to work with protein 3D structures **built using macromolecular modeling tools** and to investigate whether COSMIS score could capture constraint imposed by protein-protein oligomerization (Caffrey et al., 2004; Mintseris and Weng, 2005; Li et al., 2019), we compiled a set of 41 potassium ion channel (KCN) genes (Supplementary Table 10) for which variants have been annotated in ClinVar.

6. *Perhaps the authors could generalize this result by using RoseTTAFold to compute predicted protein-protein complexes for other proteins, or by using whatever protein-protein complexes are reported in the RoseTTA Fold paper.*

Response: We agree that the result could be generalized by using RoseTTAFold or AlphaFold to predict the structures of protein-protein complexes in human. The referenced paper (Baek et al., 2021) and two other related papers (Evans et al., 2021; Humphreys et al., 2021) present exciting new methods, but neither of them has been used to predict the structures of human protein-protein complexes at the proteome-scale. Thus, generalizing the interface results using RoseTTAFold or AlphaFold predicted protein complex structures would require a massive computational and technical effort. As an alternative, we identified and analyzed the COSMIS scores of 1,678 human homodimers from the PDB. Results from this new analysis have been incorporated into the revised manuscript. We copy the relevant sections and figures below, with new text and results highlighted in red.

Revision:

Updates in Results:

We obtained structures for these KCN proteins in their homo-oligomeric states either from the PDB or through homology modeling using the SWISS-MODEL interactive workspace (Waterhouse et al., 2018). Collectively, we structure-mapped and computed two sets of COSMIS scores, based on monomers and oligomers, respectively, for 4,762 interface and 14,331 non-interface sites in these potassium channels. As expected, we found that on average interface sites are significantly more constrained than non-interface sites (median COSMIS score -1.3 vs. -1.1, $p = 1.3 \times 10^{-16}$, two-sided Mann-Whitney U test; Fig. 8a). When computed based on oligomer structures, the scores of interface sites shift significantly to more negative values (median difference -0.13, $p = 8.1 \times 10^{-8}$, two-sided Mann-Whitney U test; Fig. 8b). **This was further confirmed by analyzing the scores of a larger set of 1,678 diverse human homodimers from the PDB (median difference -0.11, $p = 3.1 \times 10^{-12}$, two-sided Mann-Whitney U test; Methods; Supplementary Fig. 14), suggesting that COSMIS captures additional constraint on interface sites contributed by sites in neighboring subunits.**

Updates in Discussion:

Our approach nevertheless has several limitations. First, the missense burden analysis and statistical identification of constrained contact sets is highly dependent upon the number and quality of variants used as references for the standing variation dataset. Current gnomAD samples carry only an average 6.3% and 10.3% of all possible missense and synonymous variants per contact set, respectively (Supplementary Fig. 10). As larger and more diverse reference genetic variation cohorts continue to increase the number of observed variants in each gene, even more

accurate estimates of constraint on contact sets will be possible. It may also be possible to decrease the contact set distance threshold to capture more specific 3D interactions. Second, the COSMIS score does not directly consider the physicochemical severity of amino acid substitutions. While these patterns likely contribute to the patterns of observed variation and mutability, explicit consideration of the severity of amino acid substitution could improve estimates of site constraint. For instance, sites tolerant to both conservative and non-conservative substitutions are likely to be under less constraint than sites that are only tolerant to conservative substitutions. Third, while it is known that sites harboring variants with lower minor allele frequencies are likely under stronger selection pressure (Hartl, 1989), as with previous approaches (Samocha et al., 2014; Gussow et al., 2016; Lek et al., 2016; Traynelis et al., 2017; Havrilla et al., 2019; Karczewski et al., 2020), we counted the number of unique variants observed at each site and did not explicitly account for their MAFs. Incorporation MAFs and demographic structure into the formulation of scores such as the pLI (Lek et al., 2016), MTR (Traynelis et al., 2017), and COSMIS remains a promising topic. **Finally, our analysis of the COSMIS scores of protein complexes is far from comprehensive, due to the lack of a proteome-wide structural database for human protein-protein interactions. Given the rapid advances in structure prediction for protein complexes (Baek et al., 2021; Evans et al., 2021; Humphreys et al., 2021), applying COSMIS to protein complexes at proteome-scale will soon be feasible.** Together, we anticipate that our framework can be further improved in the future by including larger human variation datasets, consideration of additional amino acid properties, and accounting for MAF and **protein-protein interactions**.

Updates in Methods:

Application of COSMIS to homodimeric structures from the PDB

We obtained a set of homodimeric proteins from the Interactome INSIDER resource (Meyer et al., 2018). Briefly, we first retrieved all “highest-confidence” protein-protein interfaces in human from <http://interactomeinsider.yulab.org> (accessed in Jan. 2022). We then excluded interfaces that were not calculated from PDB structures or consist of two different protein subunits. For the remaining homodimeric interfaces, we mapped their UniProt accession numbers to PDB chains using a summary table processed by SIFTS (<https://www.ebi.ac.uk/pdbe/docs/sifts/quick.html>, pdb_chain_uniprot.tsv.gz) (Dana et al., 2019). We retained proteins that were mapped to PDB files consisting of two and only two identical subunits. This step excluded homodimeric interfaces that are part of higher-order protein complexes. In summary, we computed the COSMIS scores for 1,678 proteins (Supplementary Table 13) using both monomeric and homodimeric structures from the PDB.

Minor Concerns:

1. No discussion of gnomAD versions used

Response: We elected to use gnomAD v2.1.1., largest available at the time. We have clarified this in the main text. Also, the scores can be updated as gnomAD grows.

Revision:

The COSMIS framework maps spatial constraint on proteins in high resolution

We developed the COSMIS framework to quantify the 3D spatial constraint at each site in a protein structure by analyzing the patterns of population variant distribution in protein structures (Fig. 1 and Methods). Our framework estimates the constraint on a site of interest (index site) as the depletion of missense variants in its 3D spatial neighborhood (i.e., contact set, see definition below) compared to the number expected if sites were evolving neutrally. We quantify this as the deviation of observed count of missense variants (m_o) from the expected count (m_e , accounting for transcript and codon missense mutability), divided by the standard deviation of the expected distribution (m_σ). We designate this Z score as the COSMIS score and assign it to the index site. **We obtain the observed variant count in each contact set by mapping variants catalogued by the Genome Aggregation Database (gnomAD, v2.1.1) (Karczewski et al., 2020) onto protein structures. To obtain the expected variant count distribution, we use a procedure that simulates mutation under neutral evolution based on a 64 by 3 sequence-context-dependent mutability matrix derived using the 1000 Genomes Project variant set (Auton et al., 2015).** We also compute an empirical p-value for each COSMIS score based on the simulation procedure and the resulting expected count distribution (Methods). According to our formulation of the score, a lower value indicates a greater depletion of missense variants in the spatial neighborhood and hence lower missense variation tolerance.

2. *The authors show they predict protein constraint in oligomeric states but don't use RoseTTAFold which is supposed to be better at this. Is this an opportunity for future improvement*

Response: We agree that RoseTTAFold is state-of-the-art method for identifying residues that are constrained by protein-protein interactions (Humphreys et al., 2021). We would like to clarify that we applied COSMIS to protein oligomers to show that extra constraint can be captured for interface residues when the COSMIS scores are computed using oligomeric structures. The message is that, it is desirable to use the structure of protein complex when it is available to compute the COSMIS scores. However, we would like to emphasize that it is our future plan to update the pre-computed scores when a comprehensive structural database of human protein complexes becomes available. We have discussed this point in the revised manuscript. We copy the relevant sections and figures below, with new text and results highlighted in red.

Revision:

Our approach nevertheless has several limitations. First, the missense burden analysis and statistical identification of constrained contact sets is highly dependent upon the number and quality of variants used as references for the standing variation dataset. Current gnomAD samples carry only an average 6.3% and 10.3% of all possible missense and synonymous variants per contact set, respectively (Supplementary Fig. 16). As larger and more diverse reference genetic variation cohorts continue to increase the number of observed variants in each gene, even more accurate estimates of constraint on contact sets will be possible. It may also be possible to decrease the contact set distance threshold to capture more specific 3D interactions. Second, the COSMIS score does not directly consider the physicochemical severity of amino acid substitutions. While these patterns likely contribute to the patterns of observed variation and mutability, explicit

consideration of the severity of amino acid substitution could improve estimates of site constraint. For instance, sites tolerant to both conservative and non-conservative substitutions are likely to be under less constraint than sites that are only tolerant to conservative substitutions. Third, while it is known that sites harboring variants with lower minor allele frequencies are likely under stronger selection pressure (Hartl, 1989), as with previous approaches (Samocha et al., 2014; Gussow et al., 2016; Lek et al., 2016; Traynelis et al., 2017; Havrilla et al., 2019; Karczewski et al., 2020), we counted the number of unique variants observed at each site and did not explicitly account for their MAFs. Incorporation MAFs and demographic structure into the formulation of scores such as the pLI (Lek et al., 2016), MTR (Traynelis et al., 2017), and COSMIS remains a promising topic. **Finally, our analysis of the COSMIS scores of protein complexes is far from comprehensive, due to the lack of a proteome-wide structural database for human protein-protein interactions. Given the rapid advances in structure prediction for protein complexes (Baek et al., 2021; Evans et al., 2021; Humphreys et al., 2021), applying COSMIS to protein complexes at proteome-scale will soon be possible.** Together, we anticipate that our framework can be further improved in the future by including larger human variation datasets, consideration of additional amino acid properties, and accounting for MAF and **protein-protein interactions**.

3. *“With an average of 28.6 high-confidence constrained sites per protein”  are these constrained sites clustered together in 3D space?*

Response: We thank the reviewer for raising this interesting question. We analyzed the spatial distribution of the “high-confidence” constrained sites and found that the constrained sites in most proteins are significantly clustered. We copy the relevant sections and figures below, with new text and results highlighted in red.

Revision:

We consider sites with a COSMIS score for which the empirical p-value obtained from simulation is <0.01 as high-confidence (this is approximately equivalent to COSMIS score < -2.33). Overall, we find 313,204 sites (5.1%) with high-confidence constraint scores from 10,955 proteins (66.3%), with an average of 28.6 high-confidence constrained sites per protein (Fig. 3c, Supplementary Table 2). **These sites are generally clustered in 3D space (median ratio of average pairwise distance between high-confidence sites compared to that expected from permutations is 0.66) (Supplementary Fig. 7), suggesting that high-confidence constrained sites identified by COSMIS likely represent functionally important domains.** Overall, these findings suggest that the COSMIS score captures the depletion of missense variants in 3D structure-based contact sets resulting from varying functional constraint over protein space.

Fig S7. High-confidence constrained sites in most proteins are clustered. a) To determine whether the set of high-confidence constrained sites of a protein are clustered, we compare the average pairwise distance between these sites with that expected if the same number of sites were chosen randomly. We derived the expected average pairwise distance through 1,000 permutations based on the same structure used to compute the COSMIS scores. Dots above the diagonal represent proteins whose high-confidence sites are clustered. Those with an empirical p value < 0.01 are colored in salmon red. b) Examples of the clustering of high-confidence sites in three proteins. Residues at high-confidence sites are rendered in spheres colored according to element types (dark gray: carbon, blue: nitrogen, red: oxygen, yellow: sulfur).

4. Discussion section: “while also providing important biophysical insights into the potential functional roles of constrained sites”. Please provide a concrete example of this.

Response: In this work, we loosely used “biophysical” to describe any insights gained by inspecting COSMIS scores in the context of protein 3D structure. As the term “biophysical” may sometimes be used specifically to mean the dynamics or kinetics of molecular processes, we have replaced it by the term “structural” throughout the manuscript. We note that UBA5 as shown in Figure 4 (panels b and c) is one example illustrating the structural insights gained from COSMIS. In addition, we also analyzed the COSMIS scores of residues in the KCNQ1 tetrameric potassium ion channel, which also illustrates the concordance between low COSMIS scores and structural/functional importance. We copied the relevant sections and figures below, with new text and results highlighted in red.

Revision:

Fig S15. Mapping of COSMIS scores of residues of the KCNQ1 tetrameric potassium channel to structure highlights the clustering of functionally important regions. The voltage sensor domains (VSD) and the pore, which are responsible for sensing the changes in membrane potential and conducting potassium ion flow respectively, stand out as strongly constrained (more negative COSMIS scores) regions in KCNQ1. Regions that bind KCNE3 and calmodulin (CaM), which are believed to be essential regulators of KCNQ1 function (Sun and MacKinnon, 2020), are also strongly constrained. PDB ID of the structure: 6V01.

References:

- Auton, A., Brooks, L.D., Durbin, R.M., Garrison, E.P., Kang, H.M., Korbel, J.O., Marchini, J.L., McCarthy, S., McVean, G.A., and Abecasis, G.R. (2015). A global reference for human genetic variation. *Nature* 526, 68-74.
- Baek, M., DiMaio, F., Anishchenko, I., Dauparas, J., Ovchinnikov, S., Lee, G.R., Wang, J., Cong, Q., Kinch, L.N., Schaeffer, R.D., *et al.* (2021). Accurate prediction of protein structures and interactions using a three-track neural network. *Science* 373, 871-876.
- Caffrey, D.R., Somaroo, S., Hughes, J.D., Mintseris, J., and Huang, E.S. (2004). Are protein-protein interfaces more conserved in sequence than the rest of the protein surface? *Protein Sci* 13, 190-202.
- Dana, J.M., Gutmanas, A., Tyagi, N., Qi, G., O'Donovan, C., Martin, M., and Velankar, S. (2019). SIFTS: updated Structure Integration with Function, Taxonomy and Sequences resource allows

40-fold increase in coverage of structure-based annotations for proteins. *Nucleic Acids Res* **47**, D482-D489.

Evans, R., O'Neill, M., Pritzel, A., Antropova, N., Senior, A., Green, T., Žídek, A., Bates, R., Blackwell, S., Yim, J., *et al.* (2021). Protein complex prediction with AlphaFold-Multimer. *bioRxiv*, 2021.2010.2004.463034.

Glazer, A.M., Wada, Y., Li, B., Muhammad, A., Kalash, O.R., O'Neill, M.J., Shields, T., Hall, L., Short, L., Blair, M.A., *et al.* (2020). High-Throughput Reclassification of SCN5A Variants. *Am J Hum Genet* **107**, 111-123.

Gussow, A.B., Petrovski, S., Wang, Q., Allen, A.S., and Goldstein, D.B. (2016). The intolerance to functional genetic variation of protein domains predicts the localization of pathogenic mutations within genes. *Genome Biol* **17**, 9.

Hartl, D.L. (1989). *Principles of population genetics* / Daniel L. Hartl, Andrew G. Clark (Sunderland, Mass: Sinauer Associates).

Havrilla, J.M., Pedersen, B.S., Layer, R.M., and Quinlan, A.R. (2019). A map of constrained coding regions in the human genome. *Nat Genet* **51**, 88-95.

Humphreys, I.R., Pei, J., Baek, M., Krishnakumar, A., Anishchenko, I., Ovchinnikov, S., Zhang, J., Ness, T.J., Banjade, S., Bagde, S.R., *et al.* (2021). Computed structures of core eukaryotic protein complexes. *Science* **374**, eabm4805.

Karczewski, K.J., Francioli, L.C., Tiao, G., Cummings, B.B., Alfoldi, J., Wang, Q., Collins, R.L., Laricchia, K.M., Ganna, A., Birnbaum, D.P., *et al.* (2020). The mutational constraint spectrum quantified from variation in 141,456 humans. *Nature* **581**, 434-443.

Lek, M., Karczewski, K.J., Minikel, E.V., Samocha, K.E., Banks, E., Fennell, T., O'Donnell-Luria, A.H., Ware, J.S., Hill, A.J., Cummings, B.B., *et al.* (2016). Analysis of protein-coding genetic variation in 60,706 humans. *Nature* **536**, 285-291.

Li, B., Mendenhall, J., and Meiler, J. (2019). Interfaces Between Alpha-helical Integral Membrane Proteins: Characterization, Prediction, and Docking. *Comput Struct Biotechnol J* **17**, 699-711.

Meyer, M.J., Beltran, J.F., Liang, S., Fragoza, R., Rumack, A., Liang, J., Wei, X., and Yu, H. (2018). Interactome INSIDER: a structural interactome browser for genomic studies. *Nat Methods* **15**, 107-114.

Mintseris, J., and Weng, Z.P. (2005). Structure, function, and evolution of transient and obligate protein-protein interactions. *Proceedings of the National Academy of Sciences of the United States of America* **102**, 10930-10935.

Paludan-Muller, C., Ahlberg, G., Ghouse, J., Svendsen, J.H., Haunso, S., and Olesen, M.S. (2017). Analysis of 60 706 Exomes Questions the Role of De Novo Variants Previously Implicated in Cardiac Disease. *Circ Cardiovasc Genet* **10**.

Pejaver, V., Urresti, J., Lugo-Martinez, J., Pagel, K.A., Lin, G.N., Nam, H.J., Mort, M., Cooper, D.N., Sebat, J., Iakoucheva, L.M., *et al.* (2020). Inferring the molecular and phenotypic impact of amino acid variants with MutPred2. *Nat Commun* **11**, 5918.

Perszyk, R.E., Kristensen, A.S., Lyuboslavsky, P., and Traynelis, S.F. (2021). Three-dimensional missense tolerance ratio analysis. *Genome Res* **31**, 1447-1461.

Samocha, K.E., Robinson, E.B., Sanders, S.J., Stevens, C., Sabo, A., McGrath, L.M., Kosmicki, J.A., Rehnstrom, K., Mallick, S., Kirby, A., *et al.* (2014). A framework for the interpretation of de novo mutation in human disease. *Nat Genet* **46**, 944-950.

Shrestha, R., Fajardo, E., Gil, N., Fidelis, K., Kryshchak, A., Monastyrskyy, B., and Fiser, A. (2019). Assessing the accuracy of contact predictions in CASP13. *Proteins* 87, 1058-1068.

Silk, M., Pires, D.E.V., Rodrigues, C.H.M., D'Souza, E.N., Olshansky, M., Thorne, N., and Ascher, D.B. (2021). MTR3D: identifying regions within protein tertiary structures under purifying selection. *Nucleic Acids Res* 49, W438-W445.

Sun, J., and MacKinnon, R. (2020). Structural Basis of Human KCNQ1 Modulation and Gating. *Cell* 180, 340-347 e349.

Traynelis, J., Silk, M., Wang, Q., Berkovic, S.F., Liu, L., Ascher, D.B., Balding, D.J., and Petrovski, S. (2017). Optimizing genomic medicine in epilepsy through a gene-customized approach to missense variant interpretation. *Genome Res* 27, 1715-1729.

Waterhouse, A., Bertoni, M., Bienert, S., Studer, G., Tauriello, G., Gumienny, R., Heer, F.T., de Beer, T.A.P., Rempfer, C., Bordoli, L., *et al.* (2018). SWISS-MODEL: homology modelling of protein structures and complexes. *Nucleic Acids Research* 46, W296-W303.

REVIEWERS' COMMENTS

Reviewer #1 (Remarks to the Author):

The authors have addressed some unclear points and revised accordingly. I want to congratulate the authors for their exceptional work and suggest to go ahead.

Reviewer #2 (Remarks to the Author):

The authors have carefully responded to our concerns and have revised the manuscript to address the concerns and clarify the presentation of this important research.